# High rates of evolution preceded shifts to sex-biased gene expression in *Leucadendron*, the most sexually dimorphic angiosperms

Mathias Scharmann[1]*, Anthony G Rebelo[2], John R Pannell[1]

[1]Department of Ecology and Evolution, University of Lausanne, Lausanne, Switzerland; [2]Applied Biodiversity Research Division, South African National Biodiversity Institute, Cape Town, South Africa

**Abstract** Differences between males and females are usually more subtle in dioecious plants than animals, but strong sexual dimorphism has evolved convergently in the South African Cape plant genus *Leucadendron*. Such sexual dimorphism in leaf size is expected largely to be due to differential gene expression between the sexes. We compared patterns of gene expression in leaves among 10 *Leucadendron* species across the genus. Surprisingly, we found no positive association between sexual dimorphism in morphology and the number or the percentage of sex-biased genes (SBGs). Sex bias in most SBGs evolved recently and was species specific. We compared rates of evolutionary change in expression for genes that were sex biased in one species but unbiased in others and found that SBGs evolved faster in expression than unbiased genes. This greater rate of expression evolution of SBGs, also documented in animals, might suggest the possible role of sexual selection in the evolution of gene expression. However, our comparative analysis clearly indicates that the more rapid rate of expression evolution of SBGs predated the origin of bias, and shifts towards bias were depleted in signatures of adaptation. Our results are thus more consistent with the view that sex bias is simply freer to evolve in genes less subject to constraints in expression level.

**\*For correspondence:**
mathias.scharmann@unil.ch

**Competing interest:** The authors declare that no competing interests exist.

## Editor's evaluation

This analysis of sex-biased gene expression in ten species of the most sexually dimorphic Angiosperm genus Leucadendron will appeal broadly to researchers interested in the evolution of sexual dimorphism. By analysing the broad phylogenetic context of male vs female gene expression across species along with a comprehensive population genetics approach, the results of this study reinforce the view that expression divergence between the sexes often results from relaxed purifying selection rather than from the direct action of adaptive evolution.

## Introduction

Sexual dimorphism is common in dioecious plants, affecting a broad range of physiological (*Juvany and Munné-Bosch, 2015*), morphological (*Barrett and Hough, 2013*; *Dawson and Geber, 1999*; *Tonnabel et al., 2017*), life history (*Delph, 1999*), and defence traits (*Cornelissen and Stiling, 2005*), yet sexual dimorphism in plants tends to be less marked than it is in animals (*Barrett and Hough, 2013*; *Lloyd and Webb, 1977*; *Moore and Pannell, 2011*). The South African genus *Leucadendron* stands out as an exception to this pattern, with several of its approximately 80 dioecious species showing strong sexual dimorphism for leaves and plant architectural traits as well as inflorescences

**Figure 1.** Strong morphological sex differences of the leaves evolved repeatedly in the genus *Leucadendron*, yet sex-biased expression affects only few genes, and the transcriptomes of males are not similar between species, nor are transcriptomes of females similar between species. (**A**) Typical male (left) and female (right) shoot tips of *L. rubrum*, a wind-pollinated species with strong sexual dimorphism in leaves, stems, and inflorescence. (**B**) Left: supermatrix species tree; scale bar indicates expected number of substitutions per site. All branches showed full Shimodaira–Hasegawa-like support. Middle: schematic outlines of male (blue) and female (red) leaves, drawn to scale from a single example photograph; background shading indicates lower sexual dimorphism as light grey and higher sexual dimorphism as dark grey, as classified by the data of **Tonnabel et al., 2014**. Right: bar plots showing the percentage of male-biased (blue) and female-biased (red) genes among all expressed genes.

(**Bond and Midgley, 1988**; **Midgley, 2010**; **Rebelo, 2001**; **Tonnabel et al., 2014**; **Williams, 1972**). Dimorphism in different traits is highly correlated across species of *Leucadendron*, and reaches extremes in species such as *L. rubrum*, whose females grow leaves that are on average 13 times larger (in surface area) than male leaves, and in which males produce 75 times more inflorescences per stem (**Figure 1A**, **Bond and Midgley, 1988**).

The possibility of sexual selection occurring in plants has long been controversial, although evolutionary processes that act to increase mating success are well documented in plants (**Moore and Pannell, 2011**). The difference in leaf size and morphology between males and females of *Leucadendron* is thought to be the result of both sexual selection and differential costs of reproduction (**Barrett and Hough, 2013**; **Geber et al., 1999a**; **Harris and Pannell, 2008**; **Moore and Pannell, 2011**; **Obeso, 2002**). First, competition for siring success among males should favour the evolution of large floral displays (**Harder and Barrett, 2006**; **Moore and Pannell, 2011**). In *Leucadendron*, large flower displays are achieved via the production of finer external branches, which support allometrically smaller leaves (**Bond and Midgley, 1988**). **Bond and Maze, 1999** showed that males of *L. xanthoconus* with larger floral displays both attracted more pollinators and suffered greater mortality. These observations are consistent with the selection of large floral displays, and correspondingly smaller leaves (**Bond and Midgley, 1988**), due to sexual selection, ultimately opposed by viability selection (**Bond and Maze, 1999**).

Second, males and females of dioecious plants are subject to different costs and constraints imposed by their different reproductive strategies (**Bond and Midgley, 1988**). In *Leucadendron*, seeds are produced in inflorescences that develop into woody infructescences ('cones') that are costly in terms of photosynthates. Presumably because thicker branches are required for the mechanical support of large cones, they can also support larger leaves (**Bond and Midgley, 1988**). **Harris and Pannell, 2010** found that sexual dimorphism was more pronounced in those *Leucadendron* species in which females maintained their seeds for longer periods (often several years) in live cones, suggesting that the additional maternal costs of reproduction, attributable to increased transpiration and thus water use, further contributed divergence in leaf traits between the sexes. It thus seems clear that leaf morphology in *Leucadendron*, although fundamentally a vegetative trait, is intimately linked to both

male and female reproductive strategies and success. Interestingly, in certain species of *Leucadendron* with specialized pollinators, male and female inflorescences, and their surrounding leaves differ in attractive cues and provide the same pollinator with different rewards (*Hemborg and Bond, 2005*), pointing to a further link between reproductive structures and vegetative morphology.

The morphological and physiological differences between males and females of sexually dimorphic species are likely to depend on differences in gene expression, either because of divergence in gene content at loci linked to sex (on sex chromosomes; *Bachtrog et al., 2014*) or due to differences in gene expression (sex-biased gene expression, SBGE), at autosomal loci (*Ellegren and Parsch, 2007*; *Mank, 2009*; *Zemp et al., 2016*). However, it is also possible that sexual dimorphism develops without SBGE, as a consequence of sex-associated variation in genetic architectures, which could be widespread (*van der Bijl and Mank, 2021*). Either way, and to the extent that morphological sexual dimorphism is related to SBGE, we might expect species (and tissues) that show particularly striking differences between males and females in morphology also to be characterized by high levels of SBGE. To our knowledge, this prediction has only been tested at a phylogenetic scale in birds. Based on a sample of six galloanserine species, *Harrison et al., 2015* reported greater levels of SBGE in species with more exaggerated male traits. Given the well-established role played by sexual selection in bringing about sexual dimorphism, it would seem natural that the evolution of SBGE in cases such as that documented by *Harrison et al., 2015* might also often be driven by sexual selection. Whether such associations apply more generally is not yet known.

*Harrison et al., 2015* specifically tested for a correlation in birds between a proxy for the outcome of sexual selection (male trait exaggeration) and gene expression in reproductive tissues. Many further studies of SBGE have likewise focussed on reproductive tissues, or samples that include them (*Ellegren and Parsch, 2007*; *Grath and Parsch, 2016*; *Parsch and Ellegren, 2013*). Reproductive tissues include the gametes and are necessarily sexually dimorphic in morphology and gene expression, and they directly mediate fertilization and mating success (e.g., through sperm or pollen traits). Many of the genes transcribed in reproductive tissues will have had sex-specific functions for a very long time. Thus, one may expect that sexual selection, as well as sexual dimorphism and SBGE, are generally strongest in reproductive and less pronounced in non-reproductive tissues and traits. But sexual dimorphism in morphology and gene expression may also occur in non-reproductive tissues and traits. Such 'secondary' sex characters may reflect distinct reproductive strategies in terms of physiology and ecology that are associated with the ancient, primary differences of male and female gametes (*Lloyd and Webb, 1977*).

Sexual dimorphism in gene expression is thought to evolve via sex-differential selection, similar to morphological traits (*Barrett and Hough, 2013*; *Geber, 1999b*; *Lande, 1980*), but could also result from non-adaptive causes such as genetic drift in genes with similar fitness effects in both sexes (*Ellegren and Parsch, 2007*). The male-biased genes in reproductive tissues of many animals, but sometimes also female-biased genes, may show elevated rates of amino acid substitution and elevated rates of gene expression divergence (*Ellegren and Parsch, 2007*; *Harrison et al., 2015*; *Khaitovich et al., 2005*; *Naqvi et al., 2019*; *Ranz et al., 2003*; *Voolstra et al., 2007*). This is probably due to frequent adaptive sweeps in cases such as *Drosophila* (*Grath and Parsch, 2016*). In many other species, however, non-adaptive causes for the accelerated evolutionary rates of sex-biased genes (SBGs) seem more plausible (*Ellegren and Parsch, 2007*; *Grath and Parsch, 2016*; *Harrison et al., 2015*). In comparison to unbiased genes, SBGs may evolve under relaxed selective constraint and lower pleiotropy, while recieving the same mutational input (*Mank and Ellegren, 2009*; *Meisel, 2011*; *Dapper and Wade, 2016*; *Gershoni and Pietrokovski, 2017*). In contrast with studies on animals, studies on plants have thus far either not found elevated substitution rates for SBGs or found reduced rates (*Grath and Parsch, 2016*; *Muyle, 2019*). Furthermore, it remains generally unclear whether the higher rates found for SBGs existed already before sex-biased expression, or else, whether they changed coincidentally or after the evolution of sex bias (compare *Orr, 2000*; *Papakostas et al., 2014*). To conclude, the relative importance of adaptive versus non-adaptive processes in gene expression divergence between the sexes remains an open question.

Here, we sequenced and analysed male and female transcriptomes in fully developed leaves of 10 *Leucadendron* species sampled from across the genus. Although leaves are non-reproductive tissue, they may, as outlined before, still be submitted to sex-specific selection in *Leucadendron*. We then address two principle questions. First, we asked whether among-species variation in morphological

sexual dimorphism correlates with variation between the sexes in levels of gene expression. Second, we inferred the history of recruitment, evolution, and rate of turnover of genes with sex-biased expression across the genus to ask whether SBGs acquired their characteristic rates of evolution before or after they became dimorphic in expression levels. Our study represents the first phylogenetic analysis of the evolution of SBG expression for plants, and our results urge caution in invoking adaptive evolution as the sole reason for high rates of expression evolution observed for SBGs.

## Results and Discussion

We based our analysis on transcriptomes sequenced from species sampled from all major clades of the genus *Leucadendron*, recapturing variation generated over several tens of millions of years of evolution during the genus' radiation (*Sauquet et al., 2009*), which is reflected in up to 3.7 % sequence differences at synonymous sites between species. We specifically paired closely related species that differed strongly in their level of sexual dimorphism. We also included in our sample one individual of the hermaphroditic species *Leucospermum reflexum* as an outgroup (sequence difference to *Leucadendron* spp. approximately 5.3%).

### Levels of SBGE in *Leucadendron* leaves span the entire range reported from other dioecious plants

De novo assembly of transcriptomes resulted in 73,307–570,280 contigs per species, of which 34,901–92,840 (13%–48%) were identified as plant-derived (*Supplementary file 1* – Table S1). The abundant and diverse non-plant transcripts mainly belonged to fungi and bacteria, which is to be expected for samples taken from wild, openly growing plants. After filtering of the plant-derived transcripts, we inferred a set of 16,194 'orthogroups' and quantified their expression across the 11 plant transcriptomes. In what follows, we refer to these orthogroups as 'genes', although they are best understood as gene families rather than single genes. We tested for differential gene expression between six male and five or six female transcriptomes per species using edgeR (*Robinson et al., 2010*) following convention, we defined genes as sex biased on the basis of a 5 % false discovery rate and a minimum twofold difference in expression between the sexes (see Methods). In total, we identified 650 genes that showed SBGE in at least one species (*Supplementary file 1* – Table S2). We report further analyses in terms of these genes, although we note that exploration of more stringent (yielding fewer SBGs) and more permissive thresholds (yielding more SBGs) led to qualitatively similar conclusions (Appendix 1). The extent of SBGE was markedly different between species, ranging from a minimum of seven male-biased and three female-biased genes in *L. ericifolium* (0.1 % of expressed genes) to 138 male-biased and 141 female-biased genes in *L. dubium* (2.5 %; *Figure 1B*). Similar patterns were revealed when SBGs were counted, or when the magnitude of bias (fold-change) was cumulated over the genes (*Supplementary file 1* – Table S3).

Although *Leucadendron* includes species with some of the strongest manifestations of morphological sexual dimorphism recorded in flowering plants (*Barrett and Hough, 2013*; *Harris and Pannell, 2010*; *Midgley, 2010*), our results indicate that the genus is not exceptional in terms of SBGE in leaves, even in the most morphologically dimorphic species. For instance, our results compare with the low number of SBGE in leaves of *Populus* (*Robinson et al., 2014*) and *Salix* (*Darolti et al., 2018*; *Sanderson et al., 2019*), in which <0.1 % of genes were sex biased, as well as the somewhat higher estimates reported for leaves of *Silene latifolia* (*Zemp et al., 2016*) and vegetative tissues of *Mercurialis annua* (*Cossard et al., 2019*), which had about 2 % of SBGs. The fraction of SBGs in *Leucadendron* is also similar to that found for exclusively non-reproductive tissues in certain animals (e.g., only one gene was found to be sex biased in the spleen of birds; *Harrison et al., 2015*), but non-reproductive tissues of other animals may express much higher fractions of genes with sex bias (e.g., hundreds or thousands of SBGs were recorded across mammal species, albeit on the basis of lower stringency; *Naqvi et al., 2019*).

### SBGE is not positively correlated with sexual dimorphism in *Leucadendron* leaves

Next, we sought evidence for a positive correspondence between variation in SBGE and levels of sexual dimorphism in *Leucadendron* leaves as found for a number of traits related to the inferred

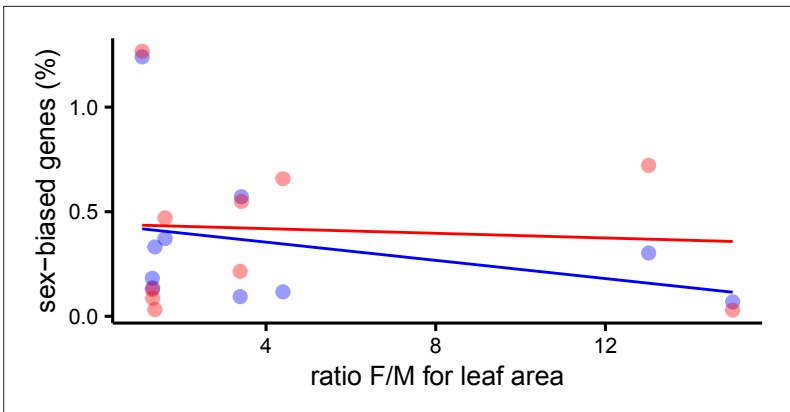

**Figure 2.** Percent of sex-biased genes in leaves as a function of morphological sexual dimorphism (ratio of female over male leaf area) in 10 species of *Leucadendron*. Male-biased expression is in blue, and female-biased expression is in red points resp. lines. The underlying data are found in **Supplementary file 1** – Table S3.

The online version of this article includes the following figure supplement(s) for figure 2:

**Figure supplement 1.** Sex-biased gene expression as a function of morphological sexual dimorphism in the leaves of 10 species of *Leucadendron*, shown in 6 alternative scatter plots for all combinations of 2 sexual dimorphism measures and 3 statistics of sex-biased gene expression.

**Figure supplement 2.** Heatmap and cluster dendrogram for biological processes putatively regulated by sex-biased genes in 10 *Leucadendron* species.

---

intensity of sexual selection in birds (*Harrison et al., 2015*). Specifically, we asked whether convergent evolution of sexual dimorphism in *Leucadendron* for leaf morphology is associated with convergence of SBGE. Our data clearly reject this hypothesis: morphological sexual dimorphism, measured either as the ratio of female/male leaf area, or specific leaf area (leaf mass per area), were never positively correlated with the number, the proportion, or the cumulative fold-changes of SBGs (phylogenetic least-squares regressions, *Figure 2* and *Figure 2—figure supplement 1*). For instance, *L. ericifolium* shows very strong leaf size dimorphism but had only seven male-biased and three female-biased genes (0.1 % of expressed genes), whereas *L. dubium* shows very little evidence for morphological dimorphism but had 138 male-biased and 141 female-biased genes (2.5%). We also failed to find any genes that were consistently sex biased in species with either low or high morphological dimorphism; instead, the identity of SBGs was largely unique in each species. Thus, we postulate that the development of dimorphic leaf size in *Leucadendron* is regulated by few genes, implying that most of the SBGs we found are not functionally related to leaf size. They might instead be related to leaf physiology, or have little functional relevance.

Annotation of *Leucadendron* SBGs against *Arabidopsis thaliana* genes revealed very diverse putative functions (*Supplementary file 1* – Tables S2, S5 and S6). While there were some obvious sex differences within species, such as male bias for phenylpropanoid/flavonoid biosynthesis and female bias for carbon assimilation in *L. dubium* (*Supplementary file 1* – Table S7), there were no clear male or female functional patterns across species (*Figure 2—figure supplement 2*). On the contrary, the two gene functional clusters most strongly overrepresented in SBGs as compared to all genes were the same among the male- and female-biased genes over all 10 species (involving flavonoid biosynthesis and secreted extracellular proteins; *Supplementary file 1* – Tables S5 and S6). This result points to the possibility that molecular sexual dimorphism could to some degree be reversed between different species.

While our analysis is based on leaf traits, we conjecture that vegetative SBGE in *Leucadendron* is also unlikely to be related to dimorphism in other traits, such as height and branching architecture. This is because dimorphism in these traits in *Leucadendron* tends to be correlated with leaf dimorphism (*Bond and Midgley, 1988*; *Harris and Pannell, 2010*), though analysis of gene expression in inflorescences would merit future study. It is also possible that variation in SBGE between *Leucadendron* species could be related to variation in genetic sex linkage and sex chromosomes, which often contain SBGs at elevated densities (*Ellegren and Parsch, 2007*; *Mank, 2009*; *Zemp et al., 2016*).

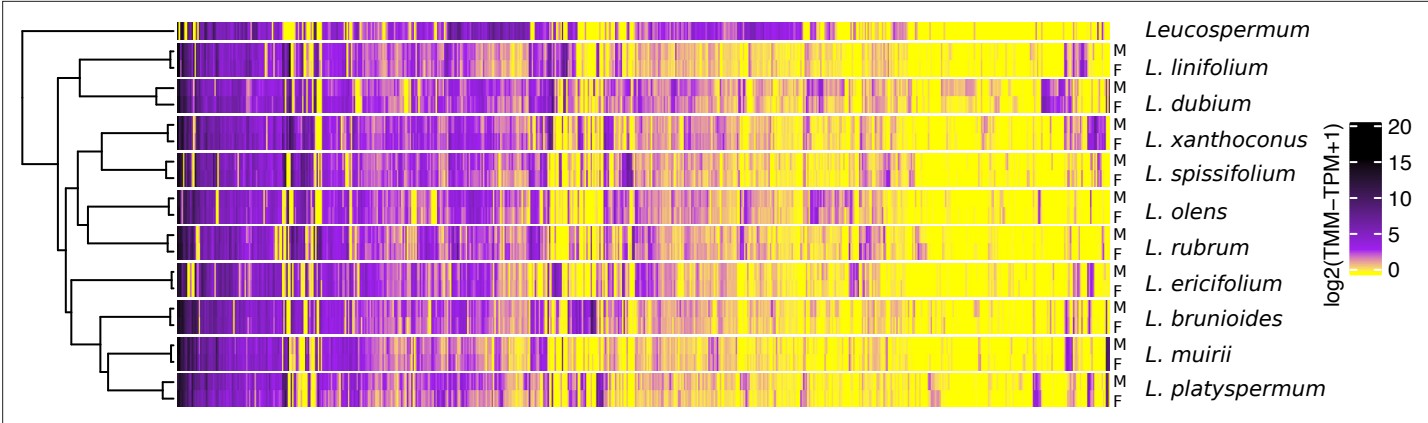

**Figure 3.** Gene expression heatmap and hierarchical clustering dendrogram for the sexes of 10 *Leucadendron* species and the hermaphroditic *Leucospermum* outgroup, for sex-biased genes only (650 genes).
Gene expression values (columns) are the mean log$_2$(TMM − TPM + 1) per species and sex. The clustering of groups (rows) is based on distances calculated as 1 − Pearson's correlation coefficient.

However, *Leucadendron* species do not have heteromorphic sex chromosomes (*Liu et al., 2006*), so this possibility seems unlikely. Indeed, SBGs in plants appear to be predominantly autosomal, even in species with large non-recombining regions such as *S. latifolia* (*Zemp et al., 2016*).

## Most SBGs are recently evolved in *Leucadendron*, and convergence in expression patterns is rare

To gain further insight into the evolution of SBGE in the *Leucadendron* radiation, we clustered species and sexes according to gross similarity in gene expression. The males and females of a given species were more similar in their expression profiles than were individuals of the same sex but of different species, that is, samples grouped by species rather than by sex (*Figure 3*). At the level of species, clustering based on gene expression failed to reflect the DNA sequence phylogeny, indicating relatively little phylogenetic inertia in expression levels. Overall, there were no clear general male-like or female-like gene expression profiles shared across the genus, and the convergence towards sexual dimorphism observed at the morphological level is not apparent at the level of gene expression. Our result that leaf transcriptomes from dioecious plants cluster by species rather than by sex is similar to results from non-reproductive tissues of animals, which also cluster by species first, as they either

**Table 1.** Summary of inferred evolutionary histories for 650 sex-biased genes in *Leucadendron*. Permutations were used to generate numbers of genes in each category expected under the null hypothesis that the identity of sex-biased genes (SBGs) is random within each species, excluding the three genes that gained sex bias only ancestrally. All observed counts were significantly different from the null. The phylogenetic patterns of shared and divergent sex-biased gene expression (SBGE) are shown in Figure 3.

| Category of sex bias | Observed | Mean of expected | p value (one-sided) |
|---|---|---|---|
| Uniquely male biased, gained at tip | 265 | 312 | $<1 \times 10^{-5}$ |
| Uniquely female biased, gained at tip | 322 | 377.5 | $<1 \times 10^{-5}$ |
| Shared male biased, single ancestral gain | 3 | NA | NA |
| Shared female biased, single ancestral gain | 1 | NA | NA |
| Shared male biased, repeatedly gained (ancestral and/or tip) | 12 | 2.4 | $<1 \times 10^{-5}$ |
| Shared female biased, repeatedly gained (ancestral and/or tip) | 19 | 3.7 | $<1 \times 10^{-5}$ |
| Divergent sex biased (gains ancestral and/or tip) | 28 | 6.1 | $<1 \times 10^{-5}$ |

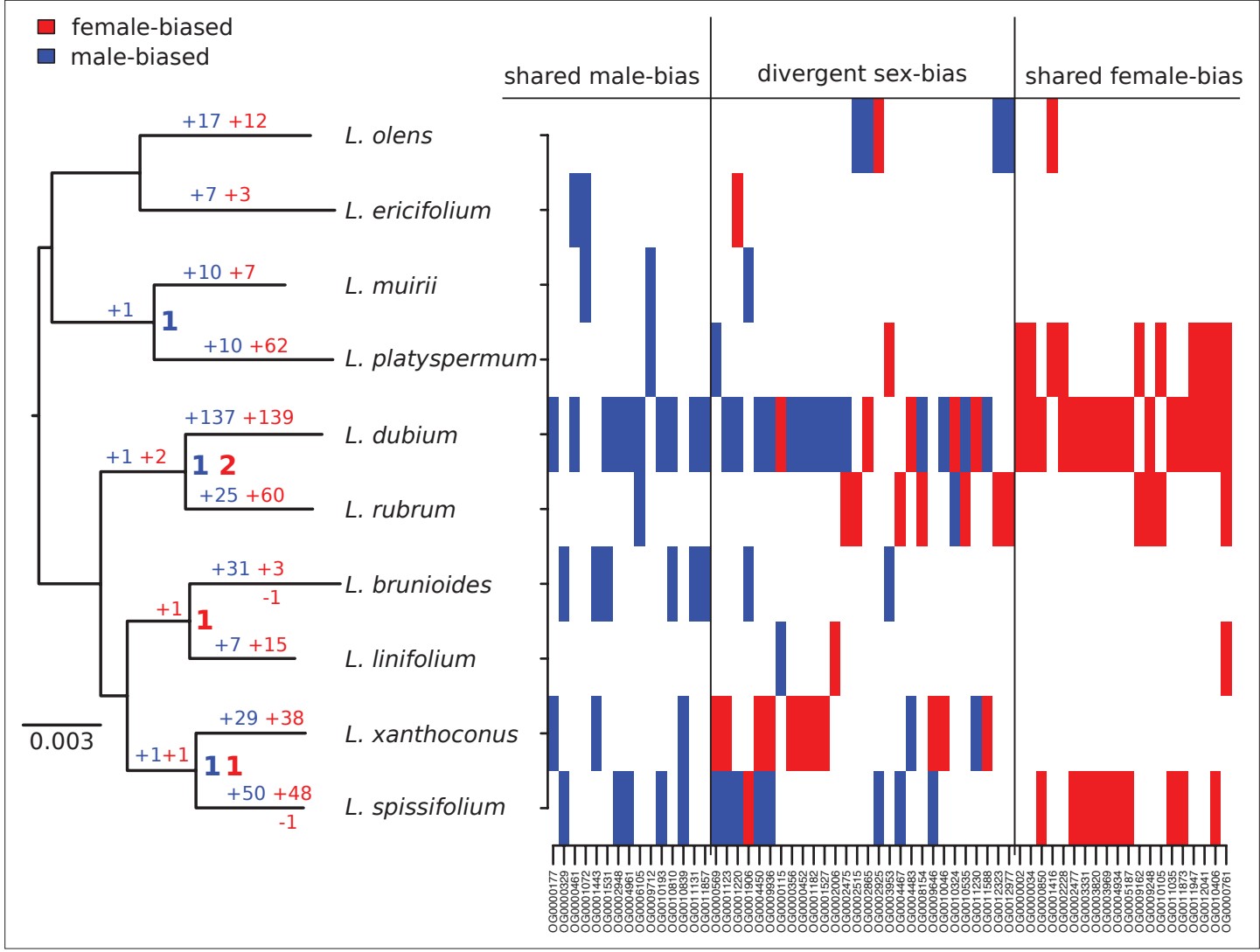

**Figure 4.** Summary of evolutionary histories inferred for sex-biased gene expression (SBGE) in *Leucadendron*.
 Left: species tree annotated with inferred numbers of sex-biased genes at ancestral nodes (bold), and gains of sex-biased genes annotated on each branch; no losses were inferred. Right: table marking the sex-bias status for the 63 genes that showed sex bias in more than 1 species (either shared bias in the same direction, or divergent sex bias). For details, see *Supplementary file 1* – Table S4.

show little overall SBGE (*Harrison et al., 2015*), or little evolutionary conservation of SBGE (*Naqvi et al., 2019*).

Our genus-wide sampling allowed us to consider the ancestral versus derived nature of SBGE for the sequenced genes, using maximum likelihood to reconstruct ancestral expression states for each SBG (*Table 1* and *Figure 4*). Of the 650 genes that showed SBGE in at least 1 species in our sample, only 63 (9.7%) were sex biased in 2 or more species; of these 63 SBGs, only 4 were likely to have been sex biased in a common ancestor. None of the recent SBGs was likely sex biased in the common ancestor of the genus (Figure 4*Figure 4*), even though dioecy is probably ancestral (*Sauquet et al., 2009*; *Tonnabel et al., 2014*). Note that this result does not imply that ancestral *Leucadendron* had little SBGE, because our reconstruction is necessarily limited to recent SBGs (see also *Harrison et al., 2015*). Together, these patterns point to a high rate of turnover of sex-biased expression among genes.

Although our results indicate that SBGE largely evolved in an idiosyncratic, lineage-specific manner in *Leucadendron*, certain genes were more likely to evolve SBGE than others. Fifty-nine out of the 63 genes showing sex bias in more than one species acquired sex bias more than once (*Table 1*, *Figure 4*, and *Supplementary file 1* – Table S4). About half of these (31 genes) acquired sex bias

of the same direction in distinct species. While convergent sex bias was limited to two species in most genes, one component of the respiratory chain (OG0010406, corresponding to *A. thaliana* gene ATMG00513) evolved female bias in three species, and another gene, putatively involved in phosphate homeostasis, evolved female bias in four species (OG0000761, SPX3, AT2G45130). The functional significance, if any, of these patterns is uncertain, but we note that phosphorous metabolism is sexually dimorphic in other dioecious plants (*Zhang et al., 2019*; *Zhang et al., 2014*). Interestingly, the remaining 28 genes with apparent convergence in SBGE actually showed reversed sex bias, that is, they evolved male bias in some species but female bias in others. While most of these genes had sex-reversed expression in only one pair of species, six genes were sex biased in opposite directions in three species (i.e., twice female biased and once male biased). Four of these genes are involved in flavonoid biosynthesis (OG0009936, OG0001220, OG0001123, and OG0004450; corresponding to AT4G22880, AT5G42800, AT1G08250, and AT1G61720), and thus have putative functions attributable to leaf pigmentation, which is often sexually dimorphic in *Leucadendron* species (*Rebelo, 2001*). The other two genes with reversed sex bias in three species are putatively involved in cytokinin biosynthesis (OG0001906 and AT4G35190), and a vacuolar sucrose invertase (OG0000569 and AT1G62660). The extent of interspecific overlap in similarly sex-biased and reversed sex-biased expression in terms of gene number was greater than expected by chance; correspondingly, the number of uniquely SBGs was lower than expected (permutation tests, all p < 10$^{-5}$; *Table 1*). It is thus clear that the set of genes recruited for SBGE during the radiation is narrower than it could have been, although not greatly.

## SBGs in *Leucadendron* do not show deviant rates of sequence evolution between species

We asked whether SBGs in *Leucadendron* differ from unbiased genes in their rates of protein evolution. In animals, SBGs often show more rapid non-synonymous sequence evolution than unbiased genes, which could reflect positive selection or lower purifying selection (*Ellegren and Parsch, 2007*; *Naqvi et al., 2019*), but previous studies of plants have so far not revealed similar tendencies (*Cossard et al., 2019*; *Muyle, 2019*; *Zemp et al., 2016*). Instead, in the only example of deviant rates of SBGs in plants so far, the male-biased genes in the inflorescences of *Salix viminalis* showed reduced rates of sequence evolution, perhaps attributable to haploid selection (*Darolti et al., 2018*). While the evolutionary rates of SBGs may be a consequence of a property correlated with sex bias, such as expression breadth over tissues and developmental stages (*Parsch and Ellegren, 2013*), it remains unclear whether sex-biased expression itself has an effect on rates of sequence evolution, or else sex-biased expression evolves more readily in genes with ancestrally and intrinsically different rates.

We first considered rates of protein evolution over long evolutionary times by estimating dN/dS between *Leucadendron* genes (majority-consensus over the whole genus) and putatively homologous genes from *A. thaliana* (which have been diverging for an estimated 130 million years; *Magallón et al., 2015*). The genes that were sex biased in at least one *Leucadendron* species showed no tendency to evolve at a rate different from those that we classified as always unbiased (*Figure 5*). Second, we considered dN/dS over the comparatively shorter divergence times between species of *Leucadendron*, using species-specific sequences. We also found no consistent effect of sex-biased expression on dN/dS in a paired assessment of genes that were observed under both sex-biased and unbiased conditions in different species (*Figure 5B and C*). Our results are therefore consistent with the previous findings for plants.

Importantly, our analysis allows us to reject the hypothesis that *Leucadendron* SBGE evolved predominantly in genes with ancestrally high or low rates of protein evolution. It also seems clear that the recent gain of SBGE did not change the evolutionary rates of affected genes. Of course, dN/dS may not provide the power to detect recent changes in evolutionary rates if few derived mutations have had the time to fix since SBGE evolved.

## Distribution of fitness effects in sex-biased genes

Analysis of population genetic polymorphism offers a perspective on sequence evolution over more recent timescales, notably via the distribution of fitness effects (DFEs) for segregating derived alleles (*Eyre-Walker and Keightley, 2007*; *Tataru et al., 2017*). To test the effect of sex-biased expression on the DFE, we focussed on one species with many SBGs (*L. dubium*, 279 SBGs), and a second species which shared none of these SBGs (*L. ericifolium*). For each species, we aligned RNA-seq reads of

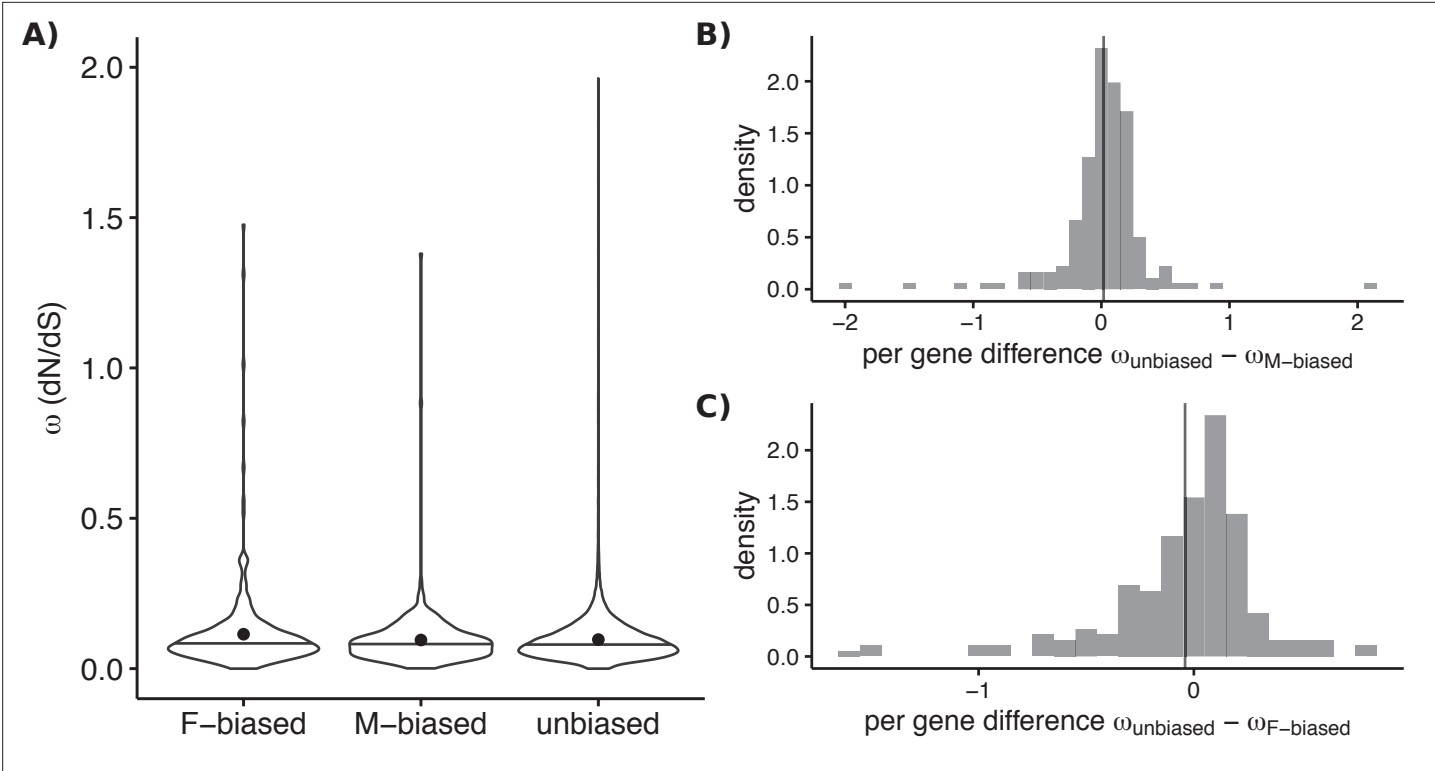

**Figure 5.** Molecular sequence evolution measured as omega (dN/dS) for sex-biased and unbiased genes of *Leucadendron*. (**A**) Omega estimated over the deep evolutionary timescale between *Arabidopsis thaliana* and the genus *Leucadendron*. Violin plots (horizontal bar marks the median, dots show the mean) for the three categories of bias status, with different sets of genes in each category. Mean omega was not significantly different between unbiased and either sex-biased category (permutation tests p > 0.05). (**B**) Omega estimated over the more recent evolutionary timescale between species of *Leucadendron*. Histograms show pairwise differences in omega for genes observed under both male bias and unbiased (i.e., in different species). The mean difference in omega is indicated by a vertical line, and does not deviate from zero significantly (permutation test, p = 0.8). (**C**) The same as (B) but for genes observed under both female bias and unbiased, again the mean difference in omega is not deviating from zero (permutation test, p = 0.9).

The online version of this article includes the following figure supplement(s) for figure 5:

**Figure supplement 1.** Distributions of fitness effects (DFEs) of sex-biased genes (SBGs; male bias, blue; female bias, red) and unbiased genes (grey) in two *Leucadendron* populations.

six males and six females, as well as two outgroup samples, to their specific transcriptome assemblies. Variants in coding regions were annotated as synonymous or non-synonymous, and polarized as derived versus ancestral. We then counted the unfolded (derived) frequency spectra of synonymous (neutral) and non-synonymous (selected) sites in genes that were unbiased in all species, and in the SBGs of *L. dubium*. In both of the two species, the SBGs of *L. dubium* contained hundreds of derived, non-synonymous polymorphisms. These data were used to fit four different DFE models with polyDFE (*Tataru and Bataillon, 2019*, see Materials and Methods for further details).

The DFE analyses revealed notable differences between SBGs and unbiased genes in *L. dubium* (*Supplementary file 1* – Table S8, *Figure 5—figure supplement 1*). The unbiased genes showed a DFE typical for genomes of outcrossing populations, with an L-shaped distribution of negative fitness effects and a small peak at positive effects (*Tataru et al., 2017*). Male-biased genes in *L. dubium* showed essentially the same DFE, except for a notable absence of adaptive polymorphisms. In contrast, female-biased genes showed fewer strongly deleterious mutations and instead more mid-level and mildly deleterious mutations. This pattern suggests that female-biased genes in *L. dubium* may experience lower selective constraints than unbiased and male-biased genes.

To further elucidate how SBGE relates to fitness effects, we compared the DFE for genes that were sex biased in *L. dubium* with the same genes that were unbiased in the distantly related *L. ericifolium*; these two *Leucadendron* species share most of their unbiased genes, but none of the

SBGs. The background DFE for genes that were unbiased in both species genes differed between the two species, with more strongly deleterious and neutral polymorphisms but fewer adaptive and mid- or mildly deleterious polymorphisms in *L. ericifolium*. This difference between species might be explained by different life histories or effective population sizes.

Importantly for the current context, the DFEs for genes that were sex biased in *L. dubium* and unbiased in *L. ericifolium* were nearly identical to the DFE for genes unbiased in both species. This may indicate that the reduced selective constraint shown by female-biased genes in *L. dubium* is the derived state. Male-biased genes of *L. dubium* seem to experience similar selection pressures, whether associated with or without sex-biased expression. Together, these results suggest that SBGE might lead to a relaxation of ancestral selective constraint, perhaps because the shift to predominantly female expression obscures these genes from selection in males. We can offer no explanation for the different DFEs of male- and female-biased genes, but if the SBGs of leaves are also expressed in gametophytes, more intense haploid selection in male- compared to female gametophytes might maintain such a selective constraint (*Beaudry et al., 2020*). Either way, our observations on DFEs must be interpreted with caution, because the confidence intervals of SBGs were wide and overlapping with those of unbiased genes, as the number of DNA sites in the site frequency spectra (SFS) of SBGs was small. Because of the low statistical power of small datasets in polyDFE, we did not attempt to

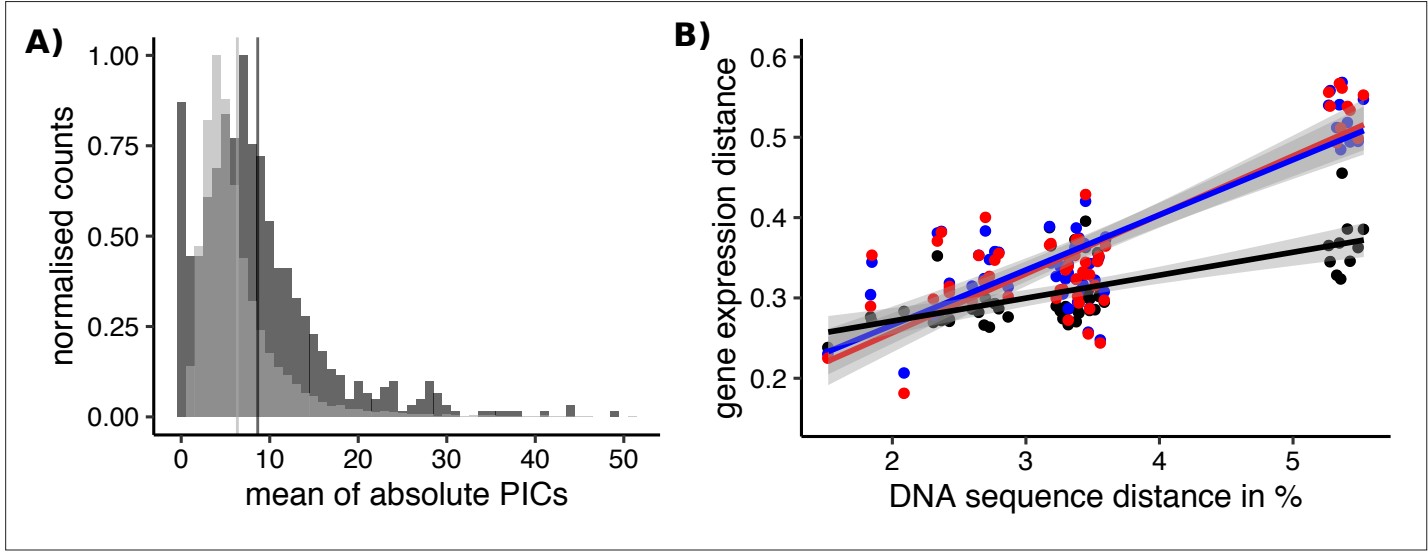

**Figure 6.** Sex-biased genes in *Leucadendron* have ancestrally and intrinsically higher rates of expression evolution.
(**A**) Histograms of mean absolute standardized phylogenetically independent contrasts (PICs) of gene expression for sex-biased (dark grey) and unbiased genes (light grey). The difference in means of the two categories is 2.3 (permutation, p = 2 × 10⁻⁵). Sex-biased expressions themselves were excluded when calculating the PICs. (**B**) Interspecific gene expression distance as a function of DNA sequence distance for 55 species pairs of dioecious *Leucadendron* and the hermaphrodite relative *Leucospermum*. Sex-biased expressions themselves were excluded when calculating gene expression distances. Gene expression distances are shown for three different categories: distance between species (mean over the sexes) for unbiased genes (black points and line), distance between males for sex-biased genes (blue points and line), and distance between females for sex-biased genes (red points and line). DNA distances > 4 % pertain to *Leucadendron–Leucospermum* pairs. The shaded envelopes around linear regression lines represent parameter standard errors. DNA sequence distance was significantly correlated with each of the three categories of gene expression distance (Mantel tests, all p ≤ 0.0225).

The online version of this article includes the following figure supplement(s) for figure 6:

**Figure supplement 1.** Demonstration that interspecific differences in expression level tend to correlate positively with the level of intraspecific expression variation, posing the risk that genes with noisy expression are mistakenly inferred to have fast rates of expression evolution.

**Figure supplement 2.** Challenging the inference of sex-biased genes by repeated permutation of the sexes.

**Figure supplement 3.** Expression specificity (Shannon entropy) over different tissues and developmental stages for sex biased and unbiased genes in *Leucadendron*, using gene expression specificity data of homologous *Arabidopsis thaliana* genes (*Klepikova et al., 2016*).

investigate the DFE of the SBGs of species other than *L. dubium*, as they contained far fewer SBGs than *L. dubium*.

## SBGs have ancestrally high rates of expression evolution in *Leucadendron*

In contrast to the absence of gross differences in the rates of sequence evolution between SBGs and unbiased genes (dN/dS), we found that SBGs have in fact been evolving more quickly than unbiased genes in terms of their expression levels (*Figure 6*). More rapid evolution of expression for SBGs has also been documented for whole-body transcriptomes of *Drosophila* (*Ranz et al., 2003*), as well as in vertebrates, in which gene expression in testes has diverged faster between species than it has in non-reproductive tissues (*Harrison et al., 2015*; *Khaitovich et al., 2005*; *Voolstra et al., 2007*). Less is known about the rates of expression evolution for SBGs in plants, but a comparison between two species of *Silene* indicated that their rates are higher than those of unbiased genes (*Zemp et al., 2016*). Our study across a diverse clade of dioecious plants adds substantially to the modest sampling in plants so far and suggests that expression evolution may be more rapid for SBGs generally, and that it evolves more quickly than the gene sequences themselves.

The phylogenetic context of our sampling allowed us to ask whether rates of expression evolution accelerated for genes once they became sex biased, or whether expression levels were already evolving rapidly before they became sex biased. The former possibility would be consistent with the outcome of sex-specific (or sexual) selection (*Hollis et al., 2014*; *Immonen et al., 2014*; *Pointer et al., 2013*; *Simmons et al., 2020*), whereas the latter possibility would be more consistent with reduced functional constraints of expression levels for SBGs, that is, they evolve more quickly because their expression levels have little effect on fitness (*Orr, 2000*; *Papakostas et al., 2014*). We thus compared rates of expression evolution between unbiased genes and SBGs. Importantly, we here inferred the rates for SBGs only among species in which these genes were actually unbiased, an analysis made possible by the largely species-specific nature of sex bias in *Leucadendron*. Phylogenetically independent contrasts (PICs; *Garland, 1992*) revealed that rates of expression evolution in *Leucadendron* tend to be higher for genes that have evolved sex bias in at least one species than for genes that were never sex biased in our sampled species (*Figure 6A*). Moreover, the interspecific expression distance of SBGs was generally greater, and increased more steeply with interspecific sequence divergence (i.e., with evolutionary time; *Figure 6B*). Because these rates of expression evolution for SBGs were inferred only from expression values in species not showing sex bias, it is clear that the elevated rates preceded these genes' acquisition of sex bias, and are independent of it.

Rates of expression evolution inferred from differences among species might appear to be higher for those genes that showed greater within-species variation in expression; indeed, intragroup variation in expression level generally correlated positively with intergroup differences (*Figure 6—figure supplement 1*, see also *Uebbing et al., 2016*). In our study, variation in expression among males or females of a given *Leucadendron* species was sometimes greater, sometimes similar, and sometimes lower for SBGs than for unbiased genes (*Supplementary file 2*). We therefore tested whether the slope of interspecific expression distance against sequence divergence in SBGs was greater than the slopes of random subsets of unbiased genes that were matched to the SBGs in their degree of intrasex expression variation. We found that all slopes of 1000 of such random but noise-matched gene subsets were lower than the slopes of the actual SBGs (*Supplementary file 3*). This pattern implies that the observed faster rate of expression evolution in *Leucadendron* is a real distinguishing feature of its SBGs and not merely an artefact of their expression noise. Moreover, permutations verified that there were more SBGs (in most species; *Figure 6—figure supplement 2*) than expected by chance alone (i.e., males and females were not exchangeable). Overall, we are thus confident that the inferred higher ancestral rates of expression evolution for SBGs are real and not an artefact of analysis or of sampling genes with especially large expression variation.

## SBGs in *Leucadendron* are ancestrally expressed in fewer tissues and stages than unbiased genes

SBGs in animals are frequently associated with higher expression specificity across a range of tissues and developmental stages (*Orr, 2000*; *Papakostas et al., 2014*), suggesting that their higher evolutionary rates may be due to reduced pleiotropic constraint and reduced phenotypic exposure to

selection (*Mank and Ellegren, 2009*; *Naqvi et al., 2019*; *Parsch and Ellegren, 2013*). We thus asked whether the SBGs in *Leucadendron* had similarly greater expression specificity, using tissue specificity in the hermaphrodite *A. thaliana* (*Klepikova et al., 2016*) and sequence similarity to *Leucadendron* to roughly estimate ancestral tissue specificity in *Leucadendron*. With this proxy, the ancestral expression specificity of SBGs in *Leucadendron* was indeed greater than that for unbiased genes (*Figure 6—figure supplement 3*). This result is consistent with the possibility that changes in the expression level of SBGs are less constrained by pleiotropic interactions than changes in the expression of unbiased genes. Because all cases of sex bias in *Leucadendron* arose recently, we infer that the lower constraints to which these genes are subject must have preceded the acquisition of sex bias, rather than be a consequence of it.

## Shifts in expression towards sex bias are depleted of signatures of adaptation

If sex-biased expression evolved in genes for which expression levels were already evolving quickly, we must consider that sex bias is in part the result of neutral evolution in gene expression (*Nourmohammad et al., 2017*), for example, due to relaxed constraint. Accordingly, we compared shifts in expression that led to sex-biased expression (sex-biased shifts) with those that did not (unbiased shifts). We defined an evolutionary shift in expression as a difference of at least 50 % between the mean expression values of sister species. (Note that sex bias is here narrowly defined for the species under scrutiny, that is, a given gene could be classed as undergoing a sex-biased shift for one species but unbiased shifts for the nine others, or not undergoing any shift at all.) We calculated for each expression shift the delta-*x* statistic (*Hsieh et al., 2003*; *Moghadam et al., 2012*; *Rifkin et al., 2003*; *Zemp et al., 2016*) to indicate whether it might have been driven by natural selection. Reflecting the notion that directional selection should both increase divergence between groups and decrease variation in the group under selection, delta-*x* is positively associated with divergence and negatively associated with polymorphism. Low expression polymorphism relative to interspecific divergence is a hallmark of adaptive evolution (*Khodursky et al., 2020*; *Nuzhdin et al., 2004*; *Ometto et al., 2011*).

We found that putatively adaptive expression shifts leading to sex bias occurred in both males and females. In particular, about 21 % of shifts leading to SBGE were putatively adaptive in at least one of the sexes, choosing the threshold of divergence five times higher than polymorphism (delta-*x* = 5.0) to class expression shifts as adaptive (*Table 2*). However, the incidence of signatures consistent with adaptation in expression levels was significantly higher among shifts not leading to sex bias (*Table 2*). This result is robust to the choice of threshold for classing expression differences between species as shifts, the choice of threshold for classing shifts as adaptive, and occurs in both increases and decreases in expression (Appendix 2). We thus reject the hypothesis that sex-biased expression

**Table 2.** Counts and chi-squared tests for shifts in gene expression in 10 *Leucadendron* spp. as classified by an index of selection and by sex bias.

Shifts are defined as differences of 1.5-fold or greater in mean expression between sister species. Delta-*x*, an index of selection, is the ratio of absolute difference in expression between sister species to the standard deviation of expression in a focal species. Shifts in the category 'high delta-*x*' (here set to 5.0 and greater) are more consistent with directional selection (adaptation) than 'low delta-*x*' shifts.

| | | Male expression shifts | | Female expression shifts | |
|---|---|---|---|---|---|
| | | Sex biased | Unbiased | Sex biased | Unbiased |
| Observed | High delta-*x* | 103 | 22,349 | 100 | 21,639 |
| | Low delta-*x* | 394 | 34,002 | 373 | 36,412 |
| Chi-squared expectations | High delta-*x* | 196.3 | 22,255.7 | 175.7 | 21,563.3 |
| | Low delta-*x* | 300.7 | 34,095.3 | 297.3 | 36,487.7 |
| Chi-squared Yates, df = 1 | | 73.134 | | 51.622 | |
| p | | $<2.2 \times 10^{-16}$ | | $6.73 \times 10^{-13}$ | |

in *Leucadendron* is generally a result of adaptive evolution. Instead, it would appear that while adaptations that establish sex-biased expression did occur, shifts towards sex-biased expression were less frequently adaptive than expression shifts unrelated to gender or sexual dimorphism.

## Concluding remarks

Our study presents the first genus-wide comparative study of the evolution of SBGE in plants. In common with the findings of previous studies of one or two species, our results confirm that SBGE in non-reproductive tissues is not as pronounced as it is in some animals. We failed to find positive correlations between SBGE in leaves and morphological dimorphism, suggesting that mature leaf transcriptomes are relatively independent from leaf morphology in function and evolution, unlike the gonad transcriptomes and morphological dimorphism of birds (*Harrison et al., 2015*; *Montgomery and Mank, 2016*). This is perhaps because the small and large leaves of *Leucadendron* males and females, respectively, mostly differ in cell number, due to timing and location of cell divisions during leaf development, but not in cell type composition and cell physiology, and because they are not under similar sex-specific selection (or they do not respond to it in a similar way). We also failed to find evidence for faster sequence evolution of SBGs compared with unbiased genes over both deep and more recent timescales, in contrast with observations for SBGs in animals. Nevertheless, standing population genetic variation in one species in our sample hints at the possibility that the gain of female-biased expression is associated with a relaxation of purifying selection. By taking a phylogenetic perspective however, we have been able to consider shifts in gene expression over evolutionary time in independent lineages with common ancestors, leading to several novel insights.

Our study has revealed an almost complete lack of convergent evolution of sex bias of individual genes, despite striking convergence in aspects of morphological dimorphism across the genus, as well as that SBGs were evidently recruited from a class of genes with intrinsically faster rates of expression evolution.

Most significantly, it is clear that the rapid rates of expression evolution for SBGs in *Leucadendron* generally predate their recent acquisition of sex bias itself, suggesting that sex bias may evolve more easily in genes that are intrinsically less constrained in relative expression level. The hypothesis of ancestrally low constraint, which is also consistent with the greater expression specificity of SBGs (in terms of tissues and developmental stages) than unbiased genes, suggests that much of the SBGE observed in plants (and perhaps animals too) may in fact have only limited functional importance. On the other hand, ratios of gene expression divergence over polymorphism suggest that at least some of the expression shifts to sex bias were nonetheless adaptive in one or both sexes. Thus, while drift in expression likely contributes to variation in gene expression and could explain much of the striking patterns of turnover among SBGs in *Leucadendron*, the potentially random walks in expression space of these genes have not been entirely invisible to the eye of sex-specific selection.

# Materials and Methods

## Plant material and RNA-seq

Based on a phylogenetic tree and analyses of trait evolution in *Leucadendron* (*Tonnabel et al., 2014*), we selected five phylogenetic pairs of species displaying low and high leaf size dimorphism. *Leucadendron* plants were sampled from wild populations in South Africa, and *L. reflexum* (outgroup) was sampled in the Botanical Garden of Zurich. We removed mature leaves just below the most recent inflorescences, cut them into pieces of <5 mm length, and immediately submerged the material in 8 ml tubes (Sarstedt 60.542.007) in ice-cold RNA-later (Ambion) or a homemade nucleic acid preservation buffer (*Camacho-Sanchez et al., 2013*). We used a manual vacuum pump to enhance infiltration of the leaves by the buffer. The samples from one species were all collected over a period of approximately 2 hrs, with males and females sampled in an alternating fashion along transects. Sample tubes were kept at 0 °C for up to 6 days and then frozen at −80 °C until further use. Total RNA was extracted from the pickled frozen leaves by grinding to a fine powder under liquid $N_2$ and purified with the Maxwell 16 LEV Plant RNA Kit (Promega Corporation, Madison, WI, USA) using a KingFisher Duo Prime robot (Thermo Fisher Scientific, Waltham, MA, USA). The RNA extracts showed generally high integrity (Bioanalyzer profiles). Sequencing libraries were constructed using the KAPA Stranded RNA-Seq kit (KAPA Biosystems, Wilmington, MA, USA) with Illumina-compatible indexed Pentadapters

(PentaBase ApS, Odense, Denmark). A total of 120 libraries were multiplexed in 6 pools of 20 each, such that each pool contained 1 male and 1 female from each species, to avoid batch effects. Each pool was sequenced in a separate lane for 150 bp paired-end reads on an Illumina HiSeq 4000 at the FGCZ. Data are deposited at the European Nucleotide Archive (ENA project PRJEB45774).

## Transcriptome de novo assemblies and identification of contaminant sequences

Adapter sequences were trimmed from the raw reads by trimmomatic (*Bolger et al., 2014*), and transcriptomes were assembled de novo for each species separately with the data of three males and three females (*Leucospermum*: one individual), using Trinity 2.5.1 (*Haas et al., 2013*). To remove contamination from epi- and endophytic microorganisms, all Trinity contigs were searched against NCBI nt (26 January 2017) using NCBI BLAST 2.7.1 (*Altschul et al., 1990*), with an *e*-value threshold of $1 \times 10^{-5}$, recording the taxonomy identifier of the best hit. Sequences were classified as 'Viridiplantae' or not, using NCBI taxonomy.

## Inference of orthogroups

We used OrthoFinder v2.3.3 (*Emms and Kelly, 2019*) to identify orthogroups, that is groups of homologous sequences descending from a common ancestral gene in our set of 11 species (i.e., including the *Leucospermum* outgroup). To this end, contigs were filtered for those classified as of Viridiplantae origin, and contig redundancy was reduced by selecting the longest isoform of each gene. On these, open reading frames (ORFs) were predicted using TransDecoder v5.5.0 (*Haas et al., 2013*), considering all ORFs >100 bp long, and shorter ORFs if they were supported by a homology search against plant proteins in SwissProt or ARAPORT11 (blastp, *e*-value cut-off $10^{-5}$). The set of predicted protein sequences from each of the 11 species was then supplied to OrthoFinder, which was run with default settings, and produced 26,553 raw orthogroups. We note that the number of Viridiplantae transcripts assembled by Trinity varied about threefold between species. This is likely mainly due to technical effects and differences in expression state of genes rather than differences in the gene content of the species' genomes, because *Leucadendron* (2$n$ = 26, *Liu et al., 2006*; as well as *Leucospermum*, 2$n$ = 24, *Rourke, 1970*) are generally diploid.

## Quantification of gene expression levels

Gene expression was quantified using pseudo-alignment with Salmon (*Patro et al., 2017*), and with the specific Trinity assembly for each species as a reference. The salmon option for metagenomic datasets ('--meta') was used, because the references contained all contaminant contigs and (partially) redundant homologous sequences. This strategy allows for maximum mapping success and avoids mis-alignment of contaminant reads to plant contigs. Using the R package tximport (*Soneson et al., 2015*), we converted Salmon abundance estimates to read counts ('scaledTPM'), and summarized to gene level, that is the OrthoFinder orthogroups plus the two additional categories 'non-coding plant transcript' and 'contaminant'. Our strategy to use all sequences and then summarize the read counts with tximport does not distinguish among orthologues, paralogs, isoforms, splice variants, or any other form of homology. Thus, the expression value of a gene in this study is a weighted summary of the expression over all contigs belonging to an orthogroup. Another consequence of the tximport summarization within orthogroups is that eventual strong differences in only some of the homologs within an orthogroup are 'diluted' in the summarized count. This would constitute a conservative bias for inference of differences in expression.

## Differential expression tests

Sex bias was tested for each species separately by differential gene expression analysis in edgeR (*Robinson et al., 2010*), with *n* = 6 per sex for *L. rubrum*, *L. spissifolium*, *L. olens*, *L. brunioides*, *L. linifolium*, *L. muirii*, *L. dubium*, and *L. xanthoconus*, and six males versus five females in *L. platyspermum* and *L. ericifolium*. For this test, 'scaledTPM' read counts were TMM-normalized (*Robinson et al., 2010*) to mitigate the possible effects of compositionality as well as library size. Within species, only genes passing a minimum expression cut-off (the count per million corresponding to more than 10 mapped reads in the smallest library) in at least 3 samples were considered as significantly expressed and included in the test; genes with zero expression in one of the sexes (sex-specific or sex-limited

genes) were thus included. Due to this filtering, only 16,194 of the 26,553 raw orthogroups were tested for differential expression in at least 1 species. Differential expression was tested with the exactTest function and tag-wise dispersion estimate. Sex bias was considered significant at a false discovery rate of 5 % (*Benjamini and Hochberg, 1995*) and a minimum twofold change between the sexes. We chose these thresholds because they are practically a standard and convention in the field of differential gene expression, and allow our results to be directly compared to other studies on SBGE. The effects of the choice of threshold on our results and conclusions are explored in Appendix 1.

## Expression data preparation for analyses other than differential expression tests

For further analyses, the expression levels of 16,194 genes were normalized for gene length, library size, and compositionality over all samples of all species together, resulting in TMM-normalized TPM values (TMM = 'Trimmed Mean of M-values', TPM = 'transcipts per million', *Robinson et al., 2010*), and transformed by $\log_2(x + 1)$. We note that this strategy included genes that had zero expression in some species. ComplexHeatmap (*Gu et al., 2016*) was used to visualize gene expression and to cluster species and sexes by gene expression distance (1 − Pearson's $r$).

## Species tree and interspecific sequence divergence

The orthogroups contained 3032 one-to-one orthologues which were present in each of the 11 taxa. These orthologue sequences were cleaned from putatively non-homologous domains using PREQUAL v1.02 (*Whelan et al., 2018*), aligned with MAFFT v7.455 (*Katoh and Standley, 2013*) on the peptide level, and finally back-translated to coding sequences. A species tree was estimated from a concatenated supermatrix using RAxML v8.2.12 (*Stamatakis, 2014*) with the GTRCAT substitution model. Uncertainty was evaluated with RAxML's Shimodaira–Hasegawa-like algorithm. The same supermatrix was used to count raw sequence divergence at fourfold degenerate sites between all pairs of taxa (custom script).

## Comparison of leaf dimorphism against sex-biased expression

We quantified sexual dimorphism of *Leucadendron* leaves as the ratio of the averages of the surface area per leaf for females over that for males (data from *Tonnabel et al., 2014*), or as the ratio of the specific leaf areas (leaf mass per leaf area). Several fresh or wet-preserved leaves per species and sex were photographed (Nikon AF-S 60 mm f/2.8 ED Micro) with a scale bar, and surface area measured in ImageJ (*Schneider et al., 2012*). Leaves were then dried and weighed on an analytical balance (Mettler Toledo XP2003SDR). These were regressed against the number, proportion, and cumulative fold-changes of male- and female-biased genes using a phylogenetic least-squares model (*Pinheiro et al., 2018*), and the species tree inferred in this study.

## Reconstruction of ancestral states

To infer where along the species tree sex-biased expression had evolved, we reconstructed the ancestral states for each SBG, with the three discrete character states 'unbiased', 'male-bias', and 'female-bias', using maximum-likelihood implemented in the ace function in the R package APE v5.3 (*Paradis and Schliep, 2019*).

## Permutation tests for number of shared SBGs

We devised a permutation procedure to test whether the observed frequencies and gene identities of repeated and divergent evolution of sex bias were different from a random expectation. Cases in which two or more species share sex bias for a gene, and if this sex bias is not due to ancestral sex bias, can be interpreted as repeated evolution and suggest sex-related functional relevance. However, if the identities (and hence putative functions) of SBGs were irrelevant, we may nevertheless expect some coincidental overlap in SBGs between species. This random expectation is analogous for genes with divergent sex bias. We generated such random expectations by permuting the identities of SBGs within each species 10,000 times while keeping the numbers intact, and scoring the interspecific overlaps. Four genes with an ancestral gain of sex-biased expression were excluded from this analysis. p

values were quantified as the proportion of overlaps in permuted datasets that were greater or equal to the observed overlaps.

## Functional gene annotation

*Leucadendron* genes were annotated against *A. thaliana* genes (Araport11.201606) by BLASTP search of the majority-consensus peptide sequence, retaining only best hits and applying an *e*-value threshold of $10^{-5}$. The corresponding *Arabidopsis* gene identifiers were used to transfer putative gene functions (*Cheng et al., 2017*) and information on gene expression specificity (*Klepikova et al., 2016*) from *Arabidopsis* to *Leucadendron*. A broad level functional categorization was generated through the TAIR web portal and the DAVID database (*Huang et al., 2009*), which was also used to cluster gene functional annotations.

## Rates of coding sequence evolution: dN/dS

We assessed dN/dS for sex-biased and unbiased genes over two contrasting evolutionary timescales. First, divergence between *A. thaliana* and *Leucadendron*, that is substitutions accumulated over c. 130 million years of separation, was estimated using the coding sequences of Araport11 genes, and the majority-consensus coding sequences of *Leucadendron* orthogroups (see 'functional gene annotation'). Pairs of *Arabidopsis* and matching *Leucadendron* sequences were pruned from putatively non-homologous domains with PREQUAL v1.02 (*Whelan et al., 2018*), aligned with MAFFT v7.455 (*Katoh and Standley, 2013*) on the peptide level, and finally back-translated to coding sequences. We then fitted PAML's model yn00 (*Yang, 2007*). We tested for differences in mean omega between unbiased, male-biased, and female-biased cases on the basis of 10,000 permutations using the function permTS in the R package perm.

Second, we estimated dN/dS over the more recent evolutionary timescale between *Leucadendron* species. To this end, we estimated dN/dS for sex-biased and unbiased genes in each species with PAML's (*Yang, 2007*) branch model 2 as implemented in the python package ete3 (*Huerta-Cepas et al., 2016*), with one foreground sequence (branch) and three background sequences, chosen as the most similar sequences from other species. Because 'genes' in our study are orthogroups, any gene may have multiple sequences per species. Thus, dN/dS was estimated for each sequence per orthogroup and species, and the average dN/dS was calculated. This estimate of omega for an orthogroup in a focal species describes how the sequences of that orthogroup, on average, evolved from the last common ancestor with the most similar sequences in different species, that is, any evolution leading towards a focal species. We then conducted a paired test for genes that were sex biased in at least one *Leucadendron* species but unbiased in at least one other *Leucadendron* species in our sample. Thus, we calculated for each gene the difference between omega under the unbiased condition and omega under the male-biased condition, or under the female-biased condition. We tested whether the average difference between sex-biased and unbiased conditions was smaller or greater than zero on the basis of 10,000 permutations as above.

## Rates of coding sequence evolution: DFEs

We analysed the DFEs between sex-biased and unbiased genes for *L. dubium*, the species in our sample with the highest number of SBGs (138 male-biased genes, 141 female-biased genes, 10,846 unbiased genes). For comparison, we also analysed DFEs in *L. ericifolium*, chosen because it is a distant relative of *L. dubium* and because most of *L. dubium*'s SBGs were also expressed in *L. ericifolium*, but none were also sex biased. We aligned RNA-seq reads of six males and six females of *L. dubium*, as well as one male *L. rubrum* and one male *L. ericifolium*, to the *L. dubium* transcriptome assembly, using bwa mem (*Li, 2013*). Likewise, we aligned RNA-seq reads of six males and six females of *L. ericifolium*, and as outgroups one male *L. platyspermum* and one male *L. rubrum*, to the *L. ericifolium* transcriptome assembly. Read alignments were filtered for primary alignments and properly paired alignments, and against supplementary alignments using samtools (*Li et al., 2009*). Invariant sites and variants (excluding indels) were called with bcftools (*Li et al., 2009*), using at most 250 reads per site and individual. The genotypes were filtered with VCFtools (*Danecek et al., 2011*) to include only biallelic SNPs and invariant sites at a minimum read depth of 3, and without missing data. Only coding regions in Viridiplantae contigs were retained (as defined by the Transdecoder structural annotation, and the taxonomy annotation, see above). Variants were annotated as synonymous or

non-synonymous by SNPeff (*Cingolani et al., 2012*). We polarized alleles as derived versus ancestral using a maximum-likelihood method (*Keightley and Jackson, 2018*), with *L. rubrum* and *L. ericifolium* serving as outgroups for *L. dubium*, and *L. platyspermum* and *L. rubrum* serving as outgroups for *L. ericifolium*. A custom script was used to count the unfolded (derived) SFS of synonymous (neutral) and non-synonymous (selected) sites, as well as the divergence against the outgroups (derived alleles fixed in the sample of the focal species). In *L. dubium*, we counted three different SFS pairs for (1) unbiased genes of *L. dubium*, (2) for male-biased genes of *L. dubium*, and (3) for female-biased genes of *L. dubium*. In *L. ericifolium*, we counted pairs of SFSs for (1) unbiased genes of *L. ericifolium*, (2) for male-biased genes of *L. dubium*, and (3) for female-biased genes of *L. dubium*. The total numbers of sequenced synonymous and non-synonymous sites, that is including variant and invariant sites, were estimated with the 'mutational opportunity method' of *Nei and Gojobori, 1986*. In both *L. dubium* and *L. ericifolium* populations, the male- and female-biased genes of *L. dubium* contained hundreds of derived, non-synonymous polymorphisms, and a few dozen fixed derived non-synonymous alleles (substitutions).

These data were used to fit DFEs with polyDFE (*Tataru et al., 2017*; *Tataru and Bataillon, 2019*). We chose the default model family ('C'), which models DFEs as a mixture of gamma and exponential distributions. We fitted these four different models: (1) a full DFE with adaptive alleles, and with ancestral allele inference error; (2) a full DFE with adaptive alleles, and no ancestral allele inference error; (3) a deleterious-only DFE, and with ancestral allele inference error; and (4) a deleterious-only DFE, and no ancestral allele inference error. All models included nuisance parameters to mitigate distortions of the allele frequency spectrum by factors such as demographic history. Divergence data were included. The fitting procedure itself was run with the basin-hopping algorithm with up to 10 iterations to avoid local maxima. Using the provided postprocessing R code, the models were ranked by the Akaike-information criterion (AIC; *Akaike, 1973*), and parameter estimates were calculated as the AIC-weighted average over all four models. We evaluated uncertainty of the parameter estimates by 200 bootstrap replicates sampled at the site level of the SFSs, to which the four different models were fitted in proportion to their AIC weight. Discretized versions of the point estimates of DFEs, together with 95 % confidence intervals from the bootstrap replicates (i.e., 2.5th and 97.5th percentiles), were plotted using ggplot2 (*Wickham et al., 2020*).

We restricted our DFE analyses to the about 140 male- or female-biased genes of *L. dubium*, because all other species contained far fewer SBGs, at most around 50 male- or female-biased genes. The CIs for bins of the discretized DFE for the SBGs of *L. dubium* were already large and spanned on average 30%, and up to 77%, of the potential parameter space. In contrast, CIs from the large sets of unbiased genes in *L. dubium* and *L. ericifolium* spanned only 5%–10% of the potential parameter space.

## Rates of gene expression evolution: PICs

The mean of the absolute standardized PICs (*Felsenstein, 1985*) per gene was employed as a measure of the rate of expression evolution (*Garland, 1992*). For each gene, PICs were calculated based on the species tree and the mean $\log_2$(TMM − TPM + 1) expression values per species, using the pic function in APE. Sex-biased expressions themselves were excluded when calculating PICs, so that the PICs only measure gene expression variation without sex bias. Significance of the mean difference between SBGs and unbiased genes in the mean absolute PICs was assessed on the basis of 10,000 permutations, using the function permTS in the R package perm v1.0 (*Fay and Shaw, 2010*).

## Rates of gene expression evolution: expression distances as a function of sequence divergence

Gene expression distances between pairs of species were calculated as 1 − Pearson's correlation coefficient, using the mean $\log_2$(TMM − TPM + 1) expression levels per species and sex. This was done for three different categories, namely for unbiased genes with the average expression levels over both sexes, for SBGs with the male expression levels, and for SBGs with the female expression levels. Note that for SBGs, this includes only expression levels and species in which these genes were not sex biased; hence sex-biased expression itself did not contribute to the gene expression distance for SBGs. For visualization of correlation trends, linear models were fitted for expression distances as a function of sequence divergence using ggplot2 (*Wickham et al., 2020*), or the lm function in R.

Significance of the correlations between pairwise distance matrices was evaluated by Mantel tests in vegan v2.5-6 (*Oksanen et al., 2019*) with 100,000 permutations.

We conducted a number of tests to verify that the higher expression evolutionary rates observed for SBGs were not spurious artefacts. First, we investigated correlations between intergroup and intragroup variation in gene expression, using simulated expression levels and real expression data. Second, we compared the mean coefficient of variation of expression counts between SBGs and unbiased genes within each species and sex using the function permTS in the R package perm, on the basis of 10,000 permutations. Third, we tested whether greater expression noise alone could explain the differences between sex-biased and unbiased genes in the linear model coefficients for gene expression distance between species as a function of sequence divergence between species. Hence, we compared the observed linear model intercept and slope estimates with artificial estimates from 1000 datasets in which the actual SBGs were replaced by an equal number of randomly chosen unbiased genes that matched the expression noise level of the true SBGs (i.e., in each artificial dataset, for each SBG, one unbiased gene was sampled that matched the SBG within 95%–105% of its coefficient of variation in expression over all species and sexes). Fourth, the empirical false-positive rate for SBGs was determined by conducting the differential gene expression analyses between the sexes (as described above) 1000 times, with datasets in which the sex was randomized (re-sampling males and females without replacement).

## Putative adaptive shifts in expression level

As an indicator of directional selection on gene expression level, we used the delta-*x* statistic, various definitions or descriptions of which were introduced by *Hsieh et al., 2003*; *Khaitovich et al., 2005*; *Moghadam et al., 2012*; *Ometto et al., 2011*; *Rifkin et al., 2003*; *Zemp et al., 2016*. Here, we define delta-*x* as the ratio of the absolute difference in mean expression between two groups (divergence) over the standard deviation of expression within a focal group (diversity/polymorphism). The rationale is analogous to indicators of selection on DNA sequence variation (*McDonald and Kreitman, 1991*), that is, directional selection on a trait (expression level) should both increase between-group divergence and decrease within-group variation, whereas traits evolving mainly under drift should show variation similar or greater than divergence. Delta-*x* was calculated on the basis of $\log_2$(TMM − TPM + 1) expression levels, separately for each gene, each *Leucadendron* species, and for males and females. Divergence was estimated as the difference between the mean expression levels in the sister species (as defined on the species tree), and diversity as the standard deviation over the six (or five) replicates of the focal species and sex. We classified divergences as quantitative evolutionary shifts if the interspecific fold-change of the means was at least 1.5. Shifts in expression showing a delta-*x* greater than 5.0 were classified as 'high delta-*x*', that is, a category more consistent with adaptive evolution. Counts of sex-biased shifts and unbiased shifts in the two categories 'high delta-*x*' and 'low delta-*x*' were compared using a 2 × 2 chi-squared test with Yates's continuity correction in R.

## Expression specificity

The gene expression atlas of *Klepikova et al., 2016* provides information on the expression specificity of *A. thaliana* genes, based on 24 different groups of tissues and developmental stages. To transfer this information to *Leucadendron* genes, we identified their most similar and putatively homologous *A. thaliana* genes by BLASTP search of the majority-consensus peptide sequence against *A. thaliana* genes (Araport11.201606, http://www.arabidopsis.org/ ; *Cheng et al., 2017*), retaining only best hits and applying an *e*-value threshold of $10^{-5}$. The original Shannon entropy values were transformed to a scale ranging from 0 (ubiquitous expression) to 1 (tissue- or stage-specific expression), as proposed by *Kryuchkova-Mostacci and Robinson-Rechavi, 2017*. The difference in mean expression specificity between SBGs and unbiased genes was assessed by 10,000 permutations using the function permTS in the R package perm.

## Acknowledgements

We thank Yves Cuenot and Dessislava Savova-Bianchi for assistance in the lab, and members of the Pannell lab, especially Jeanne Tonnabel, for advice and helpful discussions. We are grateful to Jörn Gerchen, Nora Villamil Buenrostro, Xinji Li, Michael Lenhard, Darren Parker, and Guillaume Cossard for helpful comments on an earlier version of the manuscript. Sequencing was performed

by the Functional Genomics Center Zurich (FGCZ) and bioinformatic analysis was carried out on the servers of the Division Calcul et Soutien à la Recherche (DCSR) at the University of Lausanne. The research was funded by a grant to JRP by the Swiss National Science Foundation (SNF grant number 310030_185196).

## Additional information

### Funding

| Funder | Grant reference number | Author |
| --- | --- | --- |
| Swiss National Science Foundation | 310030_185196 | John R Pannell |

The funders had no role in study design, data collection and interpretation, or the decision to submit the work for publication.

### Author contributions

Mathias Scharmann, Conceptualization, Data curation, Formal analysis, Investigation, Methodology, Software, Visualization, Writing – original draft, Writing – review and editing, sampling, field work, sampling, field work, sampling, field work; Anthony G Rebelo, Methodology, Project administration, Resources, Writing – review and editing, sampling, field work, sampling, field work, sampling, field work; John R Pannell, Conceptualization, Funding acquisition, Project administration, Resources, Supervision, Writing – original draft, Writing – review and editing, sampling, field work, sampling, field work, sampling, field work

### Author ORCIDs

Mathias Scharmann http://orcid.org/0000-0001-8523-6888
Anthony G Rebelo http://orcid.org/0000-0002-5087-262X
John R Pannell http://orcid.org/0000-0002-0098-7074

### Decision letter and Author response

Decision letter https://doi.org/10.7554/eLife.67485.sa1
Author response https://doi.org/10.7554/eLife.67485.sa2

## Additional files

### Supplementary files

• Supplementary file 1. Supplementary Tables S1 to S8.

- Table S1. Descriptive statistics of denovo assembled transcriptome sequences.

- Table S2. List of all 650 sex-biased genes and direction of sex bias per *Leucadendron* species, together with annotated AT gene identifier.
- Table S3. Percentages, counts and cumulative fold-changes of sex-biased genes, and metrics of morphological leaf dimorphism for each of the *Leucadendron* species.

- Table S4. Overview of the 63 genes that were sex biased in 2 or more species among the set of 10 *Leucadendron* species.
- Table S5. Functional annotations and annotation clusters using the default annotations at medium stringency in the DAVID database for male-biased genes in *Leucadendron*.

- Table S6. Functional annotations and annotation clusters using the default annotations at medium stringency in the DAVID database for female-biased genes in *Leucadendron*.

- Table S7. Functional annotations and annotation clusters using the default annotations at medium stringency in the DAVID database for for sex-biased genes per each of 10 *Leucadendron* species.

- • Table S8. Overview of polyDFE datasets, their model fits and parameter estimates (bins in the discretized distribution of fitness effects, DFE). df = degrees of freedom.

• Supplementary file 2. Plots showing comparisons of coefficient of variation per sex for sex-biased and unbiased genes, separately for each *Leucadendron* species.

• Supplementary file 3. Plots showing comparisons of linear model coefficients for gene expression distance between species as a function of sequence divergence between species, for sex-biased genes and unbiased genes.
 Shown are the observed intercept and slope estimates, together with histograms of the same estimates for 1000 datasets in which the actual sex-biased genes were replaced by an equal number of unbiased genes that are matched to the expression noise level of sex-biased genes (i.e., within 95%–105% of their coefficient of variation in expression over all species and sexes).

• Transparent reporting form

### Data availability

Sequencing data have been deposited at the European Nucleotide Archive (ENA project PRJEB45774).

The following dataset was generated:

| Author(s) | Year | Dataset title | Dataset URL | Database and Identifier |
|---|---|---|---|---|
| Scharmann M, Rebelo A, Pannell J | 2021 | High rates of evolution preceded shifts to sex-biased expression in Leucadendron, the most sexually dimorphic angiosperms | https://doi.org/10.5061/dryad.jsxksn0b4 | Dryad Digital Repository, 10.5061/dryad.jsxksn0b4 |
| Scharmann M, Rebelo A, Pannell J | 2021 | High rates of evolution preceded shifts to sex-biased gene expression in Leucadendron, the most sexually dimorphic angiosperms | https://www.ebi.ac.uk/ena/browser/view/PRJEB45774 | EBI, PRJEB45774 |

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

## Appendix 1

## Robustness of the results under different thresholds for defining sex-biased expression

In the main text, we present results for sex-biased genes (SBGs) as defined by differential gene expression testing in edgeR (*Robinson et al., 2010*), and application of conventional thresholds for the significance of the test and the effect size:

- *Conventional*: false discovery rate (FDR, Benjamini–Hochberg procedure; *Benjamini and Hochberg, 1995*) of 5 %; minimum twofold expression change between the sexes.

Here, we report our key results under two additional thresholds for the definition of sex-biased expression:

- *Stringent*: FDR 0.1%, minimum threefold change between the sexes.
- *Permissive*: uncorrected p value smaller or equal to 5%, no minimum fold-change between the sexes.

Source data and R code for these analyses are provided as Supporting Documents, in the subdirectories labelled SD_01 through SD_09 at https://doi.org/10.5061/dryad.jsxksn0b4 .

As expected, in comparison with the conventional thresholds, the stringent thresholds yielded much fewer SBGs, whereas the more permissive thresholds resulted in many additional SBGs (*Appendix 1—table 1*), yet among the 10 species of *Leucadendron*, the ranking of species with high or low numbers and percentages of SBGs remained intact despite the altered absolute counts of SBGs. The further patterns were also robust, with three minor exceptions:

- Using the permissive threshold, the average dN/dS estimated between *Arabidopsis* and *Leucadendron* of female-biased genes was slightly higher than dN/dS of unbiased genes. The median dN/dS of female-biased genes, however, was not different, implying that the. This might indicate that some of the genes which evolved to a weakly female-biased expression in some *Leucadendron* species are under relatively lower ancestral selective constraint.
- Using the stringent threshold, rates of expression evolution as measured by the slope of expression distance versus sequence divergence were not different between SBGs and unbiased genes, or the Mantel tests were not significant. The stringent set of SBGs is probably too small to reveal the same trend as the much larger conventional and permissive sets of SBGs.
- Using the stringent threshold, shifts to female-biased expression were not significantly depleted in high delta-*x* values, that is these shifts appeared to have similar incidences of directional selection as unbiased expression shifts. This is probably also a consequence of considering a much smaller sample of SBGs.

**Appendix 1—table 1.** Overview of key results under three thresholds for defining sex-biased expression: stringent (0.1 % FDR, minimum threefold change), conventional (5 % FDR, minimum twofold change), and permissive (uncorrected p ≤ 5%, no minimum fold-change). Statistically non-significant results are labelled as n.s. Results that are not completely consistent are shaded in grey. Subdirectory refers to the Supporting Documents at https://doi.org/10.5061/dryad.jsxksn0b4 .

| Result | Stringent | Conventional | Permissive | Subdirectory |
|---|---|---|---|---|
| Total number of SBGs | 89 | 650 | 1973 | SD_01 |
| Minimum number of SBGs per species | *L. ericifolium* (two genes, 0.02%) | *L. ericifolium* (10 genes, 0.1%) | *L. ericifolium* (358 genes, 3.5%) | SD_01 |
| Maximum number of SBGs per species | *L. dubium* (27 genes, 0.24%) | *L. dubium* (279 genes, 2.5%) | *L. dubium* (1669 genes, 15%) | SD_01 |
| Correlation between leaf dimorphism and male-biased proportion | Weak negative correlation with the ratio of leaf areas | Weak negative correlation with the ratio of leaf areas | Weak negative correlation with the ratio of leaf areas | SD_02 |
| Correlation between leaf dimorphism and female-biased proportion | n.s. | n.s. | n.s. | SD_02 |

*Appendix 1—table 1 Continued on next page*

*Appendix 1—table 1 Continued*

| Result | Stringent | Conventional | Permissive | Subdirectory |
|---|---|---|---|---|
| Hierarchical clustering by expression values of SBGs | Cluster by species, not by sex | Cluster by species, not by sex | Cluster by species, not by sex | SD_03 |
| Count (percentage) of SBGs shared by two or more species | 3 (3.4%) | 63 (9.7%) | 373 (18.9%) | SD_04 |
| Count of genes with inferred ancestral sex bias | 0 | 4 | 8 | SD_04 |
| Count of genes with inferred repeated evolution to SBGE | 3 | 59 | 365 | SD_04 |
| dN/dS of SBGs versus unbiased genes: *Arabidopsis–Leucadendron* | Difference of averages n.s. | Difference of averages n.s. | Female-biased genes show higher average dN/dS | SD_05 |
| dN/dS within *Leucadendron*: paired test of the same genes with and without sex-biased expression | On average, no difference | On average, no difference | On average, no difference | SD_05 |
| Rates of expression evolution: PICs | Elevated for SBGs | Elevated for SBGs | Elevated for SBGs | SD_06 |
| Rates of expression evolution: slope of expression distance versus sequence divergence | Slopes of SBGs and unbiased genes are similar | Elevated for SBGs | Elevated for SBGs | SD_07 |
| Expression specificity over tissues and developmental stages | Elevated for SBGs | Elevated for SBGs | Elevated for SBGs | SD_08 |
| Signatures of adaptation of expression level (shift threshold = 1.5- fold, high delta-*x*: delta-*x* ≥ 5) | Depleted in shifts to M bias, n.s. for shifts to F bias | Depleted in shifts to SBGE | Depleted in shifts to SBGE | SD_09 |

In conclusion, the exploration of various thresholds to define SBGE showed that different thresholds do not lead to contradictive patterns, although smaller samples of SBGs may reduce statistical power.

## Appendix 2

## Robustness of inference of the proportion of adaptive shifts in gene expression (delta-*x*)

We explored the effect of threshold choices on the analysis of signatures of adaptation in expression levels, as measured by the statistic delta-*x* (*Appendix 2—table 1*; *Hsieh et al., 2003*; *Moghadam et al., 2012*; *Rifkin et al., 2003*; *Zemp et al., 2016*). This descriptive statistic is a ratio of expression divergence over diversity (polymorphism) and applies to evolutionary shifts in expression between sister species. In the main text, we present results in which expression shifts are defined as at least 1.5-fold expression difference between sister species, and shifts with delta-*x* of 5.0 and greater were classified as 'high', that is more consistent with adaptation. Here, we report additional choices of thresholds for the delta-*x* analysis (*Appendix 2—table 2*).

Furthermore, it is possible that we falsely infer adaptive expression shifts (i.e. high delta-x) for genes that have nearly or completely lost expression under completely relaxed (neutral) evolution, because such genes would have very low expression polymorphism and consequently high delta-*x* values. To investigate this hypothetical bias, we partitioned our data by increases and decreases in expression, asking whether the pattern of depletion of high-delta-*x* among expression shifts towards sex bias is observed in both directions of shift. The direction of expression changes as well as the delta-x statistic were calculated separately for each sex. We found that the depletion of high-delta-x associated with sex bias is generally more pronounced among expression decreases (*Appendix 2—table 2*), as expected under the bias hypotheses. However, expression increases in males associated with sex bias are also significantly depleted in high-delta-x values. In females, the expression increases associated with sex-bias show a trend of the same direction, although this is not statistically significant. It is thus clear that the trend towards depletion of high-delta-x among shifts to sex biased expression was consistent among both increases and decreases in expression, and consistent in both cases of using either males or females to classify genes as increased or decreased in expression. Therefore, we did not distinguish decreases and increases in the delta-x analyses presented in the main text.

**Appendix 2—table 1.** Conclusions for the proportion of adaptive shifts in gene expression (delta-*x*) in shifts to sex bias and unbiased shifts under varying thresholds.

Results that are not completely consistent with the main text choice of threshold are shaded in grey. The columns 'stringent', 'conventional', and 'permissive' refer to the DGE thresholds for defining genes as sex biased (see Appendix 1). Data and code supporting this table is found in subdirectory 'SD_09' of the Supporting Documents at https://doi.org/10.5061/dryad.jsxksn0b4 .

| Shift and delta-*x* thresholds | Stringent | Conventional | Permissive |
|---|---|---|---|
| Shift threshold = 1.5- fold, high delta-*x*: delta-*x* ≥ 1.5 | Adaptations depleted in shifts to M bias, n.s. for shifts to F bias | Adaptations depleted in shifts to SBGE | Adaptations depleted in shifts to SBGE |
| Shift threshold = 1.5- fold, high delta-*x*: delta-*x* ≥ 5 | Adaptations depleted in shifts to M bias, n.s. for shifts to F bias | Adaptations depleted in shifts to SBGE | Adaptations depleted in shifts to SBGE |
| Shift threshold = 1.5- fold, high delta-*x*: delta-*x* ≥ 10 | Adaptations depleted in shifts to SBGE | Adaptations depleted in shifts to SBGE | Adaptations depleted in shifts to SBGE |
| Shift threshold = 3- fold, high delta-*x*: delta-*x* ≥ 1.5 | Adaptations depleted in shifts to M bias, n.s. for shifts to F bias | Adaptations depleted in shifts to SBGE | Adaptations depleted in shifts to SBGE |
| Shift threshold = 3- fold, high delta-*x*: delta-*x* ≥ 5 | Adaptations depleted in shifts to SBGE | Adaptations depleted in shifts to SBGE | Adaptations depleted in shifts to SBGE |
| Shift threshold = 3- fold, high delta-*x*: delta-*x* ≥ 10 | Adaptations depleted in shifts to SBGE | Adaptations depleted in shifts to SBGE | Adaptations depleted in shifts to SBGE |

**Appendix 2—table 2.** Partitioning gene expression shifts into decreases and increases in either of the sexes shows a consistent trend of depletion of high-delta-x values in shifts towards sex-biased expression as compared to unbiased shifts.

| Type of expr. change | | Observed counts | | Obs – exp for shifts to sex bias with high delta-x | Chi-squared statistic | p value |
|---|---|---|---|---|---|---|
| | | Sex biased | Unbiased | | | |
| Expression decrease in females | High delta-x | 30 | 14,416 | | | |
| | Low delta-x | 122 | 16,273 | −41.2 | 43.977 | 3.322E−11 |
| Expression increase in females | High delta-x | 70 | 7223 | | | |
| | Low delta-x | 251 | 20,139 | −14.6 | 3.2139 | 0.07302 |
| Expression decrease in males | High delta-x | 33 | 13,999 | | | |
| | Low delta-x | 119 | 12,814 | −46.1 | 55.116 | 1.14E−13 |
| Expression increase in males | High delta-x | 70 | 8350 | | | |
| | Low delta-x | 275 | 21,188 | −27.2 | 10.337 | 0.001304 |

