## [Editor Report]

This analysis of sex-biased gene expression in ten species of the most sexually dimorphic Angiosperm genus Leucadendron will appeal broadly to researchers interested in the evolution of sexual dimorphism. By analysing the broad phylogenetic context of male vs female gene expression across species along with a comprehensive population genetics approach, the results of this study reinforce the view that expression divergence between the sexes often results from relaxed purifying selection rather than from the direct action of adaptative evolution.

---

## [Decision Letter]

**Decision letter after peer review:**

Thank you for submitting your article "High rates of evolution preceded shifts to sex-biased expression in Leucadendron, the most sexually dimorphic angiosperm" for consideration by *eLife*. Your article has been reviewed by 3 peer reviewers, one of whom is a member of our Board of Reviewing Editors, and the evaluation has been overseen by Detlef Weigel as the Senior Editor. The reviewers have opted to remain anonymous.

Essential revisions:

All reviewers appreciated a competent piece of work and an important contribution. Yet there was also a consensus that several points need to be addressed and clarified. Below is a summary of the essential revisions that are required before we can recommend inclusion in *eLife*, which are detailed in the individual reviews appended.

1) The study focuses on vegetative tissues (leaves), while sexual selection is expected to be strongest for reproductive tissues (e.g. inflorescences). It is essential that the differences in sexual dimorphism and predictions in terms of sex-biased gene expression between the two types of tissues be made clearer, especially since the distinction between somatic and gonadic tissues was already critical in the Harrison et al. (2015) paper. This lack of distinction is currently quite confusing to the readers and should be clarified.

2) The threshold to define sex-biased gene expression is arbitrary. The extent to which the results presented are robust to the choice of this threshold needs to be explored.

3) In addition to the patterns of nucleotide divergence between the ten species (omega), an analysis of sequence polymorphism could be performed with the data available to evaluate the rate of adaptive evolution of genes with sex-biased expression.

4) All reviewers have also provided suggestions to improve clarity of the presentation of this tight manuscript, either by bringing back methodological details from the supplementary into the main text or pointing aspects that require more explanation. These should be carefully considered.

*Reviewer #1 (Recommendations for the authors):*

– The wording “extreme” sexual dimorphism seems too strong. There is some sexual dimorphism in leaf phenotypes, but they are still recognizably homologous structure. I would stick to the more neutral “strong” sexual dimorphism.

– The presentation is somewhat obscured by leaving most methodological details to a supplementary material. This applies to most analyses, whose presentation in the Materials and methods section could be fleshed out substantially more in the main text, such as e.g. the arbitrary thresholds used to define sex-biased genes, how exactly morphological sexual dimorphism was measured, how rates of sequence evolution were measured, how functional annotation was transferred from Leucadendron transcriptomes to *A. thaliana* genes.

– Some wordings are awkward, such as p278 : “sex-biased genes in Leucadendron are pleiotropically less constrained in their expression levels”.

*Reviewer #2 (Recommendations for the authors):*

Abstract: Line 17 "Such sexual dimorphism must largely be due to differential gene expression between the sexes". This statement seems a little strong. Why must it be? What are the alternatives?

Line 39. The authors detail sexual dimorphism in Leucadendron flowers, but this is not the tissue sampled. What degree of dimorphism is observed in leaves and how does this relate to dimorphism observed in inflorescences? Some of this is shown in Figure 1, but it would help if there was some sort of comparison presented early, along with expectations.

Line 112. I think it is quite important to clarify what tissues these comparative studies used? Were they all based on leaves?

Line 124. There is a major difference between the phenotypic measures from Harrison et al. (2015) and those used here. Specifically, Harrison et al. focused on measures associated with different types of sexual selection, the idea being that many genes in the gonad underlying gamete production were subject to sexual selection. The authors here focus on sexual dimorphism, which is related but still quite different than sexual selection. This difference needs to be explained, and it would help a great deal if the authors could clarify their expectations for how leaf dimorphism is related to sexual selection in plants. How would this be expected to result in similar or different levels of sex-biased expression or rates of evolution?

Line 226. In contrast, Darotli et al. (2018) found that sex-biased genes in catkins evolved slower than unbiased genes, likely due to haploid selection.

*Reviewer #3 (Recommendations for the authors):*

I would suggest focusing the manuscript on the finding that most SBGE is neutral (i.e. driven by drift) and making this clearer in the title and abstract. In particular, I found the second half of the abstract very hard to understand and I only understood what the authors meant after reading the entire manuscript. I think the authors need to state in the abstract that most SBG have a recent evolution and are found on the tips of the phylogeny. This makes it possible to measure expression evolutionary rates in species in which the orthologs of SBGs are not sex-biased. SBGs have ancestral high rates of expression evolution already before becoming sex-biased, which suggests drift is the main driver of SBGE evolution. In the abstract line 19-20 "Surprisingly, we found no association between sex-biased gene expression and sexual dimorphism in morphology" should rather say "number nor percentage of SBG".

In the introduction, line 38-40, it would be good to explain in more details for non specialists why a divergence in flower morphology between males and females might be counter-selected in species pollinated by specialized pollinators (the pollinator might visit only one sex…). Which should prevent the evolution of extreme sexual dimorphism in these species, contrary to wind-pollinated plants.

In the Introduction, I struggled at first to have a clear understanding of the various expectations regarding the evolution of SBGE under drift versus selection. Maybe a summary of the expectations could be presented in a table for the two hypotheses (selection and drift): expression evolutionary rates, sequence evolutionary rates, pleiotropy, % of genes undergoing adaptation (omega and delta x), convergence among species and the link with morphological dimorphism.

It is possible that the switches to SBGE are so recent that the effect on omega is not detectable. How about using methods relying on population data to infer positive selection (Eyre-Walker and Keightley, 2009; Tataru and Bataillon, 2020; Tataru, Mollion, Glémin, and Bataillon, 2017)? The authors have the data to do it (8 individuals per species). I think such an approach would yield interesting results because the authors were able to detect positive selection for some genes on expression levels with the delta-x analysis.

– Eyre-Walker, A., and Keightley, P. D. (2009). Estimating the rate of adaptive molecular evolution in the presence of slightly deleterious mutations and population size change. Molecular Biology and Evolution, 26(9), 2097-2108. doi: 10.1093/molbev/msp119

– Tataru, P., and Bataillon, T. (2020). polyDFE: Inferring the Distribution of Fitness Effects and Properties of Beneficial Mutations from Polymorphism Data. Methods in Molecular Biology (Clifton, N.J.), 2090, 125-146. doi: 10.1007/978-1-0716-0199-0_6

– Tataru, P., Mollion, M., Glémin, S., and Bataillon, T. (2017). Inference of Distribution of Fitness Effects and Proportion of Adaptive Substitutions from Polymorphism Data. Genetics, 207(3), 1103-1119. doi: 10.1534/genetics.117.300323

I wonder if differences in numbers of SBGs among species could be due to differences in their sex-linked regions? It has been shown that sex chromosomes are enriched in SBG. Therefore, if some species have a larger non-recombining region, more SBG are expected in these species. The authors do not talk about sex determinism in the genus and whether sex chromosomes are present. I suggest they include at least a discussion on this point.

How about SBGE in flowers of Leucadendron? Has it ever been studied? Does it correlate to leaf SBGE? And to flower morphological dimorphism? Maybe the authors could discuss it or suggest it as a follow-up to this study.

Maybe there were very few inferred SBGE in leaves because the timing and place of expression within leaves are more sexually dimorphic than the expression averaged over an entire leaf? For example, to make a bigger leaf, the genes driving active cell divisions might need to be expressed for a longer time at the tip of the leaf, but when the expression of these genes is averaged across the entire sampled leaf for RNA-seq, then we can't see much difference between males and females? It might be interesting to include some discussion on that point.

The manuscript is very dense and it is hard to grab a global take-home. It would help if the results/discussion were divided into subsections with titles summarizing the results found in the following paragraphs. Here is for example a suggestion but the authors should feel free to modify it as they feel:

– A surprisingly low level of SBGE in Leucadendron leaves

– SBGE is not correlated to sexual dimorphism in Leucadendron

– Most SBGs are recently evolved in Leucadendron

– Convergence in SBGE evolution across Leucadendron species

– SBG do not have accelerated sequence evolutionary rates in Leucadendron

– SBGs have ancestrally high rates of expression evolution in Leucadendron

– Few SBG evolved adaptatively in Leucadendron

– SBG are less pleiotropic than unbiased genes in Leucadendron

If I understood correctly the Materials and methods, all sex-specific genes were excluded from the study, why? How many such genes are there? Are the conclusions of the paper modified when including them?

What does TMM normalization consist in?

About the Zemp et al. 2016 citation, the delta x analysis was actually originally from this paper:

Moghadam HK, Pointer MA, Wright AE, Berlin S and Mank JE (2012) W chromosome expression responds to female-specific selection. Proc. Natl. Acad. Sci. U. S. A. 109 8207-8211

I wonder if paralogs could bias dn/ds estimates due to the analysis of transcript orthogroups?

I think Figure S3 should be in the main text because it is very interesting.

In Figure S4: how about making a paired test comparing for each gene the omega in sex-biased external (tip) branches to the omega in unbiased external (tip) branch? This would directly estimate if sex bias causes faster sequence evolution without having the issue of comparing very different genes with very different evolutionary rates to start with.

Line 76 there is an extra.

Line 97 what does c. stand for?

Line 104 it would be good to have the numbers of male and female transcriptome for each species in this paragraph.

There is very little reference to Figure 1 in the text, it could for example be called at line 130.

Line 276: sex biased *genes*?

It was unclear to me whether Table 2 "expected" line was referring to shifts in expression not leading to sex bias or to chi-square expectations. Please rename the line name in a more specific way.

Line 396-397: only *on*.

In Figure 2 how were the leaves outline drawn? Are they a single example or a mean over multiple individuals and leaves? How was the gray-scale defined? For example it is not obvious to me why L. brunioides is light gray while L. linifolium is dark gray. Maybe include a boxplot of male over female leaf length and area in the Figure? Is cone size dimorphism correlated to leaf-size dimorphism?

Please include the bootstraps in Figure 2B phylogeny.

Line 639-640: It was unclear to me what the authors meant, did they mean that there was not a shared expression pattern between all males and a different pattern shared among all females across species?

When referring to leaf size (Supplementary Materials and methods line 118), do you mean length or width?

Supplementary Materials and methods line 191: gene not genes.

Supplementary Materials and methods line 193: what does DFE stands for? Since this abbreviation is used only once, it might be better to just remove it altogether.

**[Editors' note: further revisions were suggested prior to acceptance, as described below.]**

Thank you for resubmitting your work entitled "High rates of evolution preceded shifts to sex-biased expression in Leucadendron, the most sexually dimorphic angiosperms" for further consideration by *eLife*. Your revised article has been reviewed by 2 peer reviewers and the evaluation has been overseen by Detlef Weigel as the Senior Editor, and a Reviewing Editor.

The revised manuscript has been reviewed by the same two reviewers and the Reviewing Editor. While the manuscript was changed quite substantially, several important aspects still need to be improved. At this stage we thus cannot recommend publication in *eLife*, but would be happy to consider a fully revised version convincingly addressing the following points:

– The link between sexual dimorphism, sex-biased gene expression and sexual selection still remains confusing throughout the manuscript. Reviewer #2 offered several helpful suggestions to clarify the presentation. It is essential to improve this aspect substantially, as the presentation of the results from previous studies is misleading at several places (e.g. regarding whether the number of sex-biased genes is indeed "surprising" when compared to comparable experimental strategies, e.g. whether the finding that samples cluster by species rather than by sex actually differs from previous findings, e.g. whether elevated rates of amino-acid substitutions are commonly found in animals, e.g. whether the relationship whether sexual dimorphism and gene expression differences is as univocal as implied in the text).

– A particularly confusing aspect is that the authors seem determined to point out how their results conflict with the Harrison et al. analysis, even when they do not at all. This appears to be unnecessarily confrontational, and a fairer comparison of the results with those of this study is crucially needed, in particular with regard to the important distinction between somatic vs gonadic tissues. Again, Reviewer #2 offered helpful suggestions.

– The DFE analysis that was suggested by reviewer #3 is interesting, but it remains too superficial. (1) its presentation in the text is presented in a descriptive manner that is hard to follow and the results from this analysis were somewhat excluded from the overall message of the manuscript – this should be improved. Also, the link between the different DFE between males and females and haploid selection should be made more explicit. (2) I could not understand why the analysis was performed in one species only. I understand that the number of sex-biased genes is low in most species, but this will simply decrease power of the analysis and could still be worth reporting. Alternatively, the threshold for what a "suitably high number of sex-biased genes" is should be reported (note that lines 308-313 mention that the analysis was performed "in the genus at large" but no results are reported, so it is confusing whether that was done or not) (3) More importantly, even in the case where the analysis remains focused one species only, I do not understand why the DFE analysis was not performed on the genes that are currently NOT sex-biased in the focal species but sex-biased in the other species (just like was done to show that sex biased genes in one species were already evolving quickly in the others).

– The Delta-X analysis is also interesting, but Reviewer #2 pointed out that it is essential to distinguish increases from decreases of expression, and to account for the bias introduced by comparing genes with high- vs low-expression. Also, the sentence on line 408 seems to have the reverse meaning of what it is intended to say.

– The presentation of the hypotheses in Table 1 was suggested by reviewer #3, but in its current form it is confusing. I suggest to remove and stick to the presentation of the hypotheses in the text.

– Line 367 : the idea that tissue/stage specificity is a direct measure of pleiotropy is misleading – it is very indirect. Rather say something more direct like "sex-biased genes are expressed in less tissues/stages than unbiased genes", which is closer to the actual observation.

*Reviewer #2 (Recommendations for the authors):*

I can see the authors have changed and added quite a bit to the manuscript in the revision, but unfortunately I do not find it much improved.

1. I still find the discussion of sexual dimorphism in leaf, sexual selection and SBGE very confusing. If the authors wish to invoke sexual selection in this study, they need to offer a plausible explanation for how leaf structure is subject to sexual selection. The authors are correct that the measures of sexual selection that Harrison et al. (2015) used were also cases of dimorphism, but they were chosen from the other forms of phenotypic dimorphism in birds (of which there are many) because they are well documented to result from sexual selection. To illustrate, Harrison et al. could have used such measures as nostril size, neuronal morphology or intestinal length, all of which are dimorphic, but none of which have any known association with sexual selection. Therefore, verbiage offered by the authors in their response letter and in the introduction falls short of what is needed to explain why the measures of leaf dimorphism are related to sexual selection. The simplest solution would be for them to remove mention of sexual selection from the paper, and focus in dimorphism. If this is not agreeable, then a more appropriate explanation of how leaves are sexually selected is required.

2. From Line 60. "In Leucadendron (and more generally), morphological and physiological differences between the sexes of sexually dimorphic species must ultimately trace back to differences in gene expression (sex-biased gene expression, SBGE), which enable dimorphism in spite of the largely shared genome."

I brought this up previously but perhaps wasn't direct enough. The link between SBGE and dimorphism is by no means definitive, and to my knowledge, there has been no study that demonstrated a direct relationship by which all dimorphisms are caused by either SBGE or y- or w-linked genes. In fact, van der Bijl and Mank (2021, Evolution Letters) shows that concordant changes in gene expression in males and females can result in discordant phenotypic effects, suggesting that SBGE is not necessary for dimorphism. I would therefore revise this and other similar statements to read: "In Leucadendron (and more generally), morphological and physiological differences between the sexes of sexually dimorphic species may trace back to differences in gene expression (sex-biased gene expression, SBGE), which enable dimorphism in spite of the largely shared genome."

3. Line 100 "Under the hypotheses that sex-related adaptation drives SBGE, we expect that sex-biased genes show elevated rates of amino acid substitution and faster rates of expression divergence between species (Grath and Parsch, 2016; Hollis et al., 2014; Immonen et al., 2014; Pointer et al., 2013; Simmons et al., 2020). Tests of such predictions have provided evidence consistent with this idea in many animals (Ellegren and Parsch, 2007; Grath and Parsch, 2016; Harrison et al., 2015)."

No, this is not true. Rapid rates of evolution for sex-biased genes has been recently shown in many animals to result from non-adaptive processes. For example, Harrison et al. (2015) concluded high dN/dS is due to relaxed constraint, and was more consistent with non-adaptive change. Moreover, Gershoni and Pietrokovski 2014, Nature Communications showed that many non-synonymous changes in male-biased genes in humans are mildly deleterious. Overall, where adaptive and non-adaptive scenarios have been tested separately in sex-biased genes in animals, the majority of examples are consistent with sex-bias causing a shift in the mutation-selection equilibrium (Dapper and Wade 2016 Evolution). In essence, genes that are effectively sex-limited in expression experience the mutational input from both sexes, but are only selected in one sex. This can give a false signal of positive selection for studies using more standard assumptions.

4. Line 135. How is the number of SBGs in leaf surprising? What are the numbers from other studies? Reading Darolti et al. (2018, Molecular Ecology), they used lower fold-change thresholds and only identified seven sex-biased genes in willow leaf. Are there other studies in plants that make the number observed in this paper surprisingly low? Numbers from animal somatic tissues are in this range as well – for example, Harrison et al. found just one female-biased gene using similar fold-change thresholds in the spleen, and no male-biased genes. Based on this, I would suggest the title for this section is somewhat misleading, as perhaps there are someone more SBGs than expected based on studies in comparable tissues.

5. Line 216. This bit illustrates how the comparison to Harrison et al. (2015) seems unnecessarily contradictory. Yes, it is true that in Harrison et al. gonad samples clustered first by sex, then by species, but they also reported that when they clustered expression of spleen data, it clustered by species, then sex, just like the study in review here. This underlines the fact that the leaf tissue in question here is comparable to animal somatic tissue, not gonad. The authors need to be very careful and not over-emphasize differences that are in fact entirely concordant.

6. Delta-X comparison. It is important to differentiate increases from decreases in expression in delta- comparisons, because the loss (or very low expression and functional loss) of expression for a gene is expected under completely relaxed selection, and this would give a signature of positive selection under Delta-X (large change, little variation).

Moreover, the way the authors have set up their Delta-X measure (similar to the Ometto et al. 2011 formula) will result in a conflation between highly-expressed genes and high Delta-X. This is a problem in the formulation used in the Ometto et al. (2011) analysis, and why Moghadam et al. (2012) used a somewhat more complicated formula which corrects for expression level. If the authors would like to use this formula for gene expression evolution, they should make sure that there is no relationship between expression level and Delta-X value.

7. Table 1. I am unsure how the authors reached some of these predictions. For example, I do not understand why adaptive evolution would result in greater sequence and expression evolution than non-adaptive scenarios. One could hypothesize that very few codon changes would be adaptive, leading to lower dN/dS than relaxed constraint. Similarly, if gene expression is under selection, small changes might be expected under adaptive scenarios while relaxed constraint could lead to large changes with no effect. Similarly, dN/dS would be expected to be very high in genes under relaxed constraint, and likely higher than adaptive selection. The delta-X prediction would only be true for genes experiencing increased expression, the opposite is true to decreased expression – see my comment about Delta-X above.

8. Why was the sequence polymorphism analysis not done on all species? This bit should also be discussed more thoroughly in the conclusions.

*Reviewer #3 (Recommendations for the authors):*

The authors have satisfyingly answered my previous queries.

---

## [Author Response]

Essential revisions:All reviewers appreciated a competent piece of work and an important contribution. Yet there was also a consensus that several points need to be addressed and clarified. Below is a summary of the essential revisions that are required before we can recommend inclusion in eLife, which are detailed in the individual reviews appended.1) The study focuses on vegetative tissues (leaves), while sexual selection is expected to be strongest for reproductive tissues (e.g. inflorescences). It is essential that the differences in sexual dimorphism and predictions in terms of sex-biased gene expression between the two types of tissues be made clearer, especially since the distinction between somatic and gonadic tissues was already critical in the Harrison et al. (2015) paper. This lack of distinction is currently quite confusing to the readers and should be clarified.

We have added a full paragraph to the Introduction that lays out the expected differences between reproductive and non-reproductive tissues (traits) in the intensity of sex-specific (or sexual) selection, sexual dimorphism and sex-biased gene expression. We have also taken care to state the reproductive or non-reproductive tissues in cited references.

2) The threshold to define sex-biased gene expression is arbitrary. The extent to which the results presented are robust to the choice of this threshold needs to be explored.

We added the new Appendix 1, which presents key results under very permissive (no minimum fold-change, uncorrected p-value <= 0.05) and very stringent thresholds (minimum 3-fold change, FDR <= 0.001) for the definition of sex-bias. The patterns and conclusions of our study generally are robust to the choice of threshold to define sex-biased expression. Although the threshold of 2-fold change and 5% FDR are arbitrary, we chose these because they are practically a standard and convention in the field of differential gene expression, and allow our results to be directly compared to other studies on sex-biased gene expression.

3) In addition to the patterns of nucleotide divergence between the ten species (omega), an analysis of sequence polymorphism could be performed with the data available to evaluate the rate of adaptive evolution of genes with sex-biased expression.

We conducted these additional analyses as requested, and report the new results in an additional paragraph, a new Table and a new Figure (Figure 5—figure supplement 1).

4) All reviewers have also provided suggestions to improve clarity of the presentation of this tight manuscript, either by bringing back methodological details from the supplementary into the main text or pointing aspects that require more explanation. These should be carefully considered.

We followed these suggestions as much as possible, and the main text now contains the fully detailed version of the Materials and Methods previously situated in the supplements.

Reviewer #1 (Recommendations for the authors):– The wording “extreme” sexual dimorphism seems too strong. There is some sexual dimorphism in leaf phenotypes, but they are still recognizably homologous structure. I would stick to the more neutral “strong” sexual dimorphism.

We have replaced "extreme" with "strong".

– The presentation is somewhat obscured by leaving most methodological details to a supplementary material. This applies to most analyses, whose presentation in the Materials and methods section could be fleshed out substantially more in the main text, such as e.g. the arbitrary thresholds used to define sex-biased genes, how exactly morphological sexual dimorphism was measured, how rates of sequence evolution were measured, how functional annotation was transferred from Leucadendron transcriptomes to *A. thaliana* genes.

As requested, we have replaced the brief Materials and methods section of the main text with the fully detailed version from the former Supplementary text.

– Some wordings are awkward, such as p278 : “sex-biased genes in Leucadendron are pleiotropically less constrained in their expression levels”.

We have reworded this sentence: "This result is consistent with the possibility that changes in expression level of sex-biased genes are less constrained by pleiotropic interactions than changes in expression of unbiased genes."

Reviewer #2 (Recommendations for the authors):Abstract: Line 17 "Such sexual dimorphism must largely be due to differential gene expression between the sexes". This statement seems a little strong. Why must it be? What are the alternatives?

An alternative explanation could be sex-limited genes on sex chromosomes. We have toned down this statement, and the Introduction explains this point in greater detail.

Line 39. The authors detail sexual dimorphism in Leucadendron flowers, but this is not the tissue sampled. What degree of dimorphism is observed in leaves and how does this relate to dimorphism observed in inflorescences? Some of this is shown in Figure 1, but it would help if there was some sort of comparison presented early, along with expectations.

In the Introduction, we present a brief overview of known sexually dimorphic traits in *Leucadendro*n, including both vegetative and reproductive organs. Leaf dimorphism, the focus of our study, is mentioned first (two sentences before), and (now explicitly) later again at the end of the paragraph. We added some examples of degrees of dimorphism in leaves and inflorescences; these traits are very strongly correlated in Leucadendron (as per the data of Bond and Midgley, 1988) We have also introduced a new paragraph detailing the differences that we expect between studies of reproductive traits (tissues) and non-reproductive traits (tissues).

Line 112. I think it is quite important to clarify what tissues these comparative studies used? Were they all based on leaves?

Corrected: We now refer to "leaves and plant architecture", and we removed the cited study Bond and Maze (1999) here because it investigated dimorphism of flower traits.

Line 124. There is a major difference between the phenotypic measures from Harrison et al. (2015) and those used here. Specifically, Harrison et al. focused on measures associated with different types of sexual selection, the idea being that many genes in the gonad underlying gamete production were subject to sexual selection. The authors here focus on sexual dimorphism, which is related but still quite different than sexual selection. This difference needs to be explained, and it would help a great deal if the authors could clarify their expectations for how leaf dimorphism is related to sexual selection in plants. How would this be expected to result in similar or different levels of sex-biased expression or rates of evolution?

We added a new paragraph in the Introduction to clarify the key differences between Harrison et al.s' (2015) reasoning and ours, and the expectations how leaf sexual dimorphism could be related to sexual selection and sex-biased gene expression. We argue that sexual selection is but one of several components of sex-specific selection promoting sexual dimorphism in vegetative organs of plants. The phenotypic measures in Harrison et al. (2015) are "sexual ornamentation (dichromatism, elongated feathers, wattles, caruncles, etc.; SI Methods) as a proxy for precopulatory sexual selection". As sexual selection is well established as a driver of sexual ornamentation in birds, and it is positively correlated to sex-biased gene expression, it appears that sex-biased gene expression is also driven by sexual selection. Harrison et al.’s measures are clearly measures of sexual dimorphism (in non-reproductive organs) rather than direct measures of sexual selection. Our study, too, uses sexual dimorphism measured in non-reproductive organs (in leaf area, and leaf mass per leaf area), which should be directly comparable. However, we acknowledge that sexual dimorphism in birds and in plants cannot come about by the exact same processes, but we maintain that the specific processes and expectations are generally comparable.

Line 226. In contrast, Darotli et al. (2018) found that sex-biased genes in catkins evolved slower than unbiased genes, likely due to haploid selection.

We now present this relevant aspect in the first paragraph of the section "Origin and evolutionary rates of sex-biased genes in *Leucadendron"*, and later cite the reference again when presenting our newly added result that the distribution of fitness effects in male biased genes of L. dubium is deleterious-only.

Reviewer #3 (Recommendations for the authors):I would suggest focusing the manuscript on the finding that most SBGE is neutral (i.e. driven by drift) and making this clearer in the title and abstract. In particular, I found the second half of the abstract very hard to understand and I only understood what the authors meant after reading the entire manuscript. I think the authors need to state in the abstract that most SBG have a recent evolution and are found on the tips of the phylogeny. This makes it possible to measure expression evolutionary rates in species in which the orthologs of SBGs are not sex-biased. SBGs have ancestral high rates of expression evolution already before becoming sex-biased, which suggests drift is the main driver of SBGE evolution. In the abstract line 19-20 "Surprisingly, we found no association between sex-biased gene expression and sexual dimorphism in morphology" should rather say "number nor percentage of SBG".

We followed this helpful advice and have altered the abstract accordingly.

In the introduction, line 38-40, it would be good to explain in more details for non specialists why a divergence in flower morphology between males and females might be counter-selected in species pollinated by specialized pollinators (the pollinator might visit only one sex…). Which should prevent the evolution of extreme sexual dimorphism in these species, contrary to wind-pollinated plants.

We did not follow this suggestion, because we think there is not enough space in this already long manuscript to justify the discussion of this particular hypothesis just for the sake of completeness. The role of this paragraph is to present examples and hypothetical factors behind strong sexual dimorphism in the genus. Although the question for limitations to sexual dimorphism is logically closely related to that topic, we believe their discussion would not help the understanding of our particular study here.

In the Introduction, I struggled at first to have a clear understanding of the various expectations regarding the evolution of SBGE under drift versus selection. Maybe a summary of the expectations could be presented in a table for the two hypotheses (selection and drift): expression evolutionary rates, sequence evolutionary rates, pleiotropy, % of genes undergoing adaptation (omega and delta x), convergence among species and the link with morphological dimorphism.

We have added a table in the Introduction to summarise the expectations under different scenarios for the evolution of sex-biased genes under drift or directional selection (adaptation).

It is possible that the switches to SBGE are so recent that the effect on omega is not detectable. How about using methods relying on population data to infer positive selection (Eyre-Walker and Keightley, 2009; Tataru and Bataillon, 2020; Tataru, Mollion, Glémin, and Bataillon, 2017)? The authors have the data to do it (8 individuals per species). I think such an approach would yield interesting results because the authors were able to detect positive selection for some genes on expression levels with the delta-x analysis.– Eyre-Walker, A., and Keightley, P. D. (2009). Estimating the rate of adaptive molecular evolution in the presence of slightly deleterious mutations and population size change. Molecular Biology and Evolution, 26(9), 2097-2108. doi: 10.1093/molbev/msp119– Tataru, P., and Bataillon, T. (2020). polyDFE: Inferring the Distribution of Fitness Effects and Properties of Beneficial Mutations from Polymorphism Data. Methods in Molecular Biology (Clifton, N.J.), 2090, 125-146. doi: 10.1007/978-1-0716-0199-0_6– Tataru, P., Mollion, M., Glémin, S., and Bataillon, T. (2017). Inference of Distribution of Fitness Effects and Proportion of Adaptive Substitutions from Polymorphism Data. Genetics, 207(3), 1103-1119. doi: 10.1534/genetics.117.300323

We thank the reviewer for pointing out these very relevant analyses. Because most species had only few SBGs, we used only *L. dubium* for this, and indeed we found that in this population (species), SBGs showed different rates of adaptive evolution than unbiased genes. These new results have been added to the study in the form of a new paragraph, a new Table, and a new Figure (Figure 5—figure supplement 1). Because this is restricted to one species, these new results can, however, not change the larger picture that elevated rates of sequence evolution of SBGs are not apparent in our study.

I wonder if differences in numbers of SBGs among species could be due to differences in their sex-linked regions? It has been shown that sex chromosomes are enriched in SBG. Therefore, if some species have a larger non-recombining region, more SBG are expected in these species. The authors do not talk about sex determinism in the genus and whether sex chromosomes are present. I suggest they include at least a discussion on this point.

We added a brief discussion of this point at the end of the section "Correspondence in sexual dimorphism between morphology and gene expression"

How about SBGE in flowers of Leucadendron? Has it ever been studied? Does it correlate to leaf SBGE? And to flower morphological dimorphism? Maybe the authors could discuss it or suggest it as a follow-up to this study.

We added one sentence in the section " SBGE is not positively correlated with sexual dimorphism in *Leucadendron* leaves" to highlight the possibility that SBGE is more pronounced and more correlated with morphological dimorphism in inflorescences, which have, however, not yet been studied.

Maybe there were very few inferred SBGE in leaves because the timing and place of expression within leaves are more sexually dimorphic than the expression averaged over an entire leaf? For example, to make a bigger leaf, the genes driving active cell divisions might need to be expressed for a longer time at the tip of the leaf, but when the expression of these genes is averaged across the entire sampled leaf for RNA-seq, then we can't see much difference between males and females? It might be interesting to include some discussion on that point.

We have added this thought about the timing and location of cell divisions to our discussion of possible reasons for the lack of the correlation between leaf size dimorphism and SBGE, in the section "Concluding remarks".

The manuscript is very dense and it is hard to grab a global take-home. It would help if the results/discussion were divided into subsections with titles summarizing the results found in the following paragraphs. Here is for example a suggestion but the authors should feel free to modify it as they feel:– A surprisingly low level of SBGE in Leucadendron leaves– SBGE is not correlated to sexual dimorphism in Leucadendron– Most SBGs are recently evolved in Leucadendron– Convergence in SBGE evolution across Leucadendron species– SBG do not have accelerated sequence evolutionary rates in Leucadendron– SBGs have ancestrally high rates of expression evolution in Leucadendron– Few SBG evolved adaptatively in Leucadendron– SBG are less pleiotropic than unbiased genes in Leucadendron

We implemented these summarizing sub-titles with slight modifications in the revised Results and Discussion section.

If I understood correctly the Materials and methods, all sex-specific genes were excluded from the study, why? How many such genes are there? Are the conclusions of the paper modified when including them?

We did not exclude genes with sex-specific (sex-limited) expression and added one sentence to highlight this important point in the Materials and methods. The Supplementary Materials and Methods of the original submission already stated that such genes were not excluded.

What does TMM normalization consist in?

In the methods section, we added the explanation that TMM normalisation aims to mitigate the possible effects of compositionality as well as the library size in RNA-seq data, and we cite the study that introduced it. TMM ("trimmed-mean of M-values") is the standard method of data normalization in one of the most widely used statistical frameworks for DGE testing, edgeR. Interested readers can study all details in the reference provided.

About the Zmp et al. 2016 citation, the delta x analysis was actually originally from this paper:Moghadam HK, Pointer MA, Wright AE, Berlin S and Mank JE (2012) W chromosome expression responds to female-specific selection. Proc. Natl. Acad. Sci. U. S. A. 109 8207-8211

We here specifically used Zemp's version/definition of the statistic, but the reference Moghadam et al. 2011 was added to the main text. We now give a (non-comprehensive) list of studies that have described or defined this statistic, not always under the name "delta-x": Hsieh et al., 2003; Khaitovich et al., 2005; Moghadam et al., 2012; Ometto et al., 2011; Rifkin et al., 2003; Zemp et al., 2016. We initially chose to cite only the most recent examples of this long tradition.

I wonder if paralogs could bias dn/ds estimates due to the analysis of transcript orthogroups?

Indeed, we cannot exclude the possibility that we sometimes estimate dN/dS between paralogs rather than between strict orthologs. However, this mishap can not confound our comparison of average dN/dS in sex-biased and unbiased genes, because the incidence of this mishap should be similar in our samples of sex-biased and unbiased genes. In any case, we apply a widely used orthology inference method, and furthermore select sequences for the estimation of dN/dS strictly from different species. As described in the now extended Methods section, we calculate dN/dS after selecting a subset of sequences from an orthogroup, namely a focal sequence and the three most similar from three further species. We certainly never use two sequences from the same species.

I think Figure S3 should be in the main text because it is very interesting.

We brought back Figure S3 as the new Figure 2 to the main text.

In Figure S4: how about making a paired test comparing for each gene the omega in sex-biased external (tip) branches to the omega in unbiased external (tip) branch? This would directly estimate if sex bias causes faster sequence evolution without having the issue of comparing very different genes with very different evolutionary rates to start with.

We thank the reviewer for this reasonable suggestion. We collected for each sex-biased gene the omega under male-bias, female-bias and unbiased conditions. This implies that the omega estimates under the different conditions are from different species. For genes that showed a given condition in more than one species, we averaged the omega values across species. It turns out that the mean over all genes of the differences between omega under unbiased and male-biased conditions per gene is not different from zero. The same result is obtained for the female-biased condition. Thus, the pairwise test per gene is consistent with the original result based on comparisons between different sets of genes. There is no trend in the rates of sequence evolution between sex-biased and unbiased conditions. The new paired test data and results were added to Figure S4 and the SI Methods.

Line 76 there is an extra.

Removed.

Line 97 what does c. stand for?

Changed to "approximately".

Line 104 it would be good to have the numbers of male and female transcriptome for each species in this paragraph.

Added.

There is very little reference to Figure 1 in the text, it could for example be called at line 130.

The referred paragraph is not well illustrated by Figure 1, because the Figure does not show our test of the correlation between morphology and SBGE. No changes made.

Line 276: sex biased genes?

Yes, corrected.

It was unclear to me whether Table 2 "expected" line was referring to shifts in expression not leading to sex bias or to chi-square expectations. Please rename the line name in a more specific way.

Added to the Table: These are the 2x2 chi-squared expectations.

Line 396-397: only on.

Corrected.

In Figure 2 how were the leaves outline drawn? Are they a single example or a mean over multiple individuals and leaves? How was the gray-scale defined? For example it is not obvious to me why L. brunioides is light gray while L. linifolium is dark gray. Maybe include a boxplot of male over female leaf length and area in the Figure? Is cone size dimorphism correlated to leaf-size dimorphism?

We amended the Figure legend (Figure 1 B) to clarify that the leaf outlines were drawn as single examples from photographs. The grey scale describes the assignment as a species with either "high" or "low" sexual dimorphism, as classified by the data collected in Tonnabel et al. (2014). We do not discuss cone size dimorphism here, because this is not the focus of our manuscript, and because such data are not consistently available.

Please include the bootstraps in Figure 2B phylogeny.

The figure legend was amended: "All branches showed full Shimodaira-Hasegawa-like support." This is a type of likelihood-ratio test for nearest-neighbor interchanges and is a computationally faster equivalent of bootstrapping.

Line 639-640: It was unclear to me what the authors meant, did they mean that there was not a shared expression pattern between all males and a different pattern shared among all females across species?

Reworded for clarity: "… and the transcriptomes of males are not similar between species, nor are transcriptomes of females similar between species."

When referring to leaf size (Supplementary Materials and methods line 118), do you mean length or width?

Clarified (leaf size is here the average surface area per leaf). Please note that the former Supplementary Matherials and Methods no longer exists because it was integrated into the revised main article, corresponding to *eLife*'s specifications.

Supplementary Materials and methods line 191: gene not genes.

Corrected.

Supplementary Materials and methods line 193: what does DFE stands for? Since this abbreviation is used only once, it might be better to just remove it altogether.

The abbreviation "DGE" was replaced by "differential gene expression".

**[Editors' note: further revisions were suggested prior to acceptance, as described below.]**

The revised manuscript has been reviewed by the same two reviewers and the Reviewing Editor. While the manuscript was changed quite substantially, several important aspects still need to be improved. At this stage we thus cannot recommend publication in eLife, but would be happy to consider a fully revised version convincingly addressing the following points:– The link between sexual dimorphism, sex-biased gene expression and sexual selection still remains confusing throughout the manuscript. Reviewer #2 offered several helpful suggestions to clarify the presentation. It is essential to improve this aspect substantially, as the presentation of the results from previous studies is misleading at several places (e.g. regarding whether the number of sex-biased genes is indeed "surprising" when compared to comparable experimental strategies, e.g. whether the finding that samples cluster by species rather than by sex actually differs from previous findings, e.g. whether elevated rates of amino-acid substitutions are commonly found in animals, e.g. whether the relationship whether sexual dimorphism and gene expression differences is as univocal as implied in the text).

We have re-written and amended the section of the Introduction that presents hypotheses and evidence for the evolution of sexually dimorphic vegetative traits in *Leucadendron* by sex-specific selection. Furthermore, we made substantial revisions to correctly present previous studies. This concerns especially the causes of elevated evolutionary rates of sex-biased genes in animals, following the important suggestions of R#2. The statement about "surprisingly" low proportions of SBGs has been replaced by a more appropriate statement that reflects previous studies in a more balanced way. On the matter of transcriptomes clustering by species or rather by sex, we have removed the confusing sentence and instead now state that our observation is similar to reports from animal non-reproductive tissue. We now more carefully express our expectation that sexual dimorphism *may* be based on sex-biased gene expression.

– A particularly confusing aspect is that the authors seem determined to point out how their results conflict with the Harrison et al. analysis, even when they do not at all. This appears to be unnecessarily confrontational, and a fairer comparison of the results with those of this study is crucially needed, in particular with regard to the important distinction between somatic vs gonadic tissues. Again, Reviewer #2 offered helpful suggestions.

We have revised multiple statements in our manuscript in this regard, taking care to not mis-represent previous studies, and to avoid statements of conflict between Harrison et al. (2015) and our study. There is no conflict between the studies; our findings are similar in many aspects, except for the lack of a positive correlation between SBGE and sexual dimorphism in our results.

– The DFE analysis that was suggested by reviewer #3 is interesting, but it remains too superficial. (1) its presentation in the text is presented in a descriptive manner that is hard to follow and the results from this analysis were somewhat excluded from the overall message of the manuscript – this should be improved. Also, the link between the different DFE between males and females and haploid selection should be made more explicit.

In the course of the revisions, we have completely re-done the DFE analyses, to present the uncertainty of the estimates, and to improve their integration in this study as a whole. We have also shifted the focus of the presentation away from the single parameter α (fraction of adaptive amnio acid substitutions) to the full distribution of fitness effects, and dedicated a new section to the DFE analyses ("Distribution of fitness effects in sex-biased genes"). This re-make of the analysis has led to somewhat different DFEs, due to better data filtering and incorporation of model uncertainty (we previously presented point estimates under the best model only). We now also compare DFEs for the same genes with and without sex-biased expression, in the same spirit as other analyses in our study. The discussion of possible involvement of haploid selection was made more explicit, and a statement about the DFE analyses / population genetic polymorphism has been added to the Concluding Remarks.

2) I could not understand why the analysis was performed in one species only. I understand that the number of sex-biased genes is low in most species, but this will simply decrease power of the analysis and could still be worth reporting. Alternatively, the threshold for what a "suitably high number of sex-biased genes" is should be reported (note that lines 308-313 mention that the analysis was performed "in the genus at large" but no results are reported, so it is confusing whether that was done or not).

We explain in the reply to a similar comment by R#2 the problematic lack of statistical power for small datasets in polyDFE in detail. The newly provided bootstrap confidence intervals (CIs) for bins of the discretized DFE revealed great uncertainty already with the about 140 male- or female-biased genes of the species with most SBGs in our sample. The CIs for unbiased genes (about 6 Mb of data) spanned about 5-10% of the potential parameter space, whereas the CIs for the same parameters for SBGs spanned on average 30%, and up to 77%, of the potential parameter space (see the new DFE Figure, Figure 5—figure supplement 1). We decided that analyses based on even smaller datasets, with at most about 50 male- or female-biased genes, would yield estimates that are too uncertain to be worth reporting. We explain this briefly in the Results and Discussion, and in greater detail in the Methods section.

The formulation of "suitably high number of genes" was removed, and now the threshold numbers of around 140 (deemed just enough), and about 50 (deemed too few) are clearly reported in the Methods section.

The confusing statement in former lines 308-313 has also been removed.

3) More importantly, even in the case where the analysis remains focused one species only, I do not understand why the DFE analysis was not performed on the genes that are currently NOT sex-biased in the focal species but sex-biased in the other species (just like was done to show that sex biased genes in one species were already evolving quickly in the others).

We followed this good idea, which provided additional insights that have been added to the text. However, we reasoned that it would be a stronger result to have estimates of the DFEs for the same genes under both sex-biased and unbiased expression. Hence, we fitted DFEs to *L. dubium's* SBGs in a distantly related species (*L. ericifolium*), where these genes showed unbiased expression. Please see the new section "Distribution of fitness effects in sex-biased genes" for the new results.

– The Delta-X analysis is also interesting, but Reviewer #2 pointed out that it is essential to distinguish increases from decreases of expression, and to account for the bias introduced by comparing genes with high- vs low-expression. Also, the sentence on line 408 seems to have the reverse meaning of what it is intended to say.

We have explored these hypothetical biases pointed out by Reviewer #2, and address them below. The results for delta-x with increases and decreases considered separately are presented in a new section of Appendix 2. We corrected the confusing sentence on former line 408.

– The presentation of the hypotheses in Table 1 was suggested by reviewer #3, but in its current form it is confusing. I suggest to remove and stick to the presentation of the hypotheses in the text.

We removed Table 1 and instead present hypotheses of molecular evolution in the text, as suggested.

– Line 367 : the idea that tissue/stage specificity is a direct measure of pleiotropy is misleading – it is very indirect. Rather say something more direct like "sex-biased genes are expressed in less tissues/stages than unbiased genes", which is closer to the actual observation.

Changed as suggested.

Reviewer #2 (Recommendations for the authors):I can see the authors have changed and added quite a bit to the manuscript in the revision, but unfortunately I do not find it much improved.1. I still find the discussion of sexual dimorphism in leaf, sexual selection and SBGE very confusing. If the authors wish to invoke sexual selection in this study, they need to offer a plausible explanation for how leaf structure is subject to sexual selection. The authors are correct that the measures of sexual selection that Harrison et al. (2015) used were also cases of dimorphism, but they were chosen from the other forms of phenotypic dimorphism in birds (of which there are many) because they are well documented to result from sexual selection. To illustrate, Harrison et al. could have used such measures as nostril size, neuronal morphology or intestinal length, all of which are dimorphic, but none of which have any known association with sexual selection. Therefore, verbiage offered by the authors in their response letter and in the introduction falls short of what is needed to explain why the measures of leaf dimorphism are related to sexual selection. The simplest solution would be for them to remove mention of sexual selection from the paper, and focus in dimorphism. If this is not agreeable, then a more appropriate explanation of how leaves are sexually selected is required.

We have completely re-written and amended the second paragraph of the Introduction on hypotheses of how sexual selection indirectly acts on leaf morphology (and other non-reproductive traits). It should now be clear that we do not claim that leaves directly affect mating success, but rather indirectly. For *Leucadendron*, hypotheses of how sexual selection indirectly acts on leaf morphology (and other non-reproductive traits) fall into two broad classes, which we now describe explicitly and in greater detail. Although evidence for any of these hypotheses is to some extent circumstantial, we believe that they are plausible. Testing these hypotheses is beyond the scope of our study.

The possibility of sexual selection occurring in plants has long been controversial, and in our case it may be even more controversial as we are investigating non-reproductive organs. In the first paragraph of our article, we refer the reader to an assay on the topic of sexual selection in plants (Moore and Pannell, 2011 Current Biology). We adopt that relatively loose definition of sexual selection here ("a process that acts to increase mating success").

2. From Line 60. "In Leucadendron (and more generally), morphological and physiological differences between the sexes of sexually dimorphic species must ultimately trace back to differences in gene expression (sex-biased gene expression, SBGE), which enable dimorphism in spite of the largely shared genome."I brought this up previously but perhaps wasn't direct enough. The link between SBGE and dimorphism is by no means definitive, and to my knowledge, there has been no study that demonstrated a direct relationship by which all dimorphisms are caused by either SBGE or y- or w-linked genes. In fact, van der Bijl and Mank (2021, Evolution Letters) shows that concordant changes in gene expression in males and females can result in discordant phenotypic effects, suggesting that SBGE is not necessary for dimorphism. I would therefore revise this and other similar statements to read: "In Leucadendron (and more generally), morphological and physiological differences between the sexes of sexually dimorphic species may trace back to differences in gene expression (sex-biased gene expression, SBGE), which enable dimorphism in spite of the largely shared genome."

We revised this statement to express our expectation more cautiously; it now reads: "The morphological and physiological differences between males and females of sexually dimorphic species are likely to depend on differences in gene expression […]". Nevertheless, we would follow the explanation offered by van der Bijl and Mank (2021) that discordant phenotypic effects (sexual dimorphism) of concordant (unbiased) changes in gene expression are due to sex-specific genetic architectures, which could be (citing them) "the result of interactions with sex-biased genes in the same regulatory network, or of a sex bias in the size of the cell populations expressing the gene" (van der Bijl and Mank, 2021). This statement suggests that sexual dimorphism can result from unbiased expression changes if sex-biased expression or sexual dimorphism already exist in the background.

3. Line 100 "Under the hypotheses that sex-related adaptation drives SBGE, we expect that sex-biased genes show elevated rates of amino acid substitution and faster rates of expression divergence between species (Grath and Parsch, 2016; Hollis et al., 2014; Immonen et al., 2014; Pointer et al., 2013; Simmons et al., 2020). Tests of such predictions have provided evidence consistent with this idea in many animals (Ellegren and Parsch, 2007; Grath and Parsch, 2016; Harrison et al., 2015)."No, this is not true. Rapid rates of evolution for sex-biased genes has been recently shown in many animals to result from non-adaptive processes. For example, Harrison et al. (2015) concluded high dN/dS is due to relaxed constraint, and was more consistent with non-adaptive change. Moreover, Gershoni and Pietrokovski 2014, Nature Communications showed that many non-synonymous changes in male-biased genes in humans are mildly deleterious. Overall, where adaptive and non-adaptive scenarios have been tested separately in sex-biased genes in animals, the majority of examples are consistent with sex-bias causing a shift in the mutation-selection equilibrium (Dapper and Wade 2016 Evolution). In essence, genes that are effectively sex-limited in expression experience the mutational input from both sexes, but are only selected in one sex. This can give a false signal of positive selection for studies using more standard assumptions.

We thank the reviewer for this insight; we certainly do not wish to mis-represent previous studies. This paragraph was re-written with particular emphasis on the common conclusion that non-adaptive processes are, in many cases, more consistent with patterns of molecular evolution in sex-biased genes. We incorporated the references suggested by the reviewer, replacing Gershoni and Pietrokovski (2014) with a more comprehensive analysis by the same authors from 2017. Similar statements in our manuscript were also revised to avoid any misleading statements.

4. Line 135. How is the number of SBGs in leaf surprising? What are the numbers from other studies? Reading Darolti et al. (2018, Molecular Ecology), they used lower fold-change thresholds and only identified seven sex-biased genes in willow leaf. Are there other studies in plants that make the number observed in this paper surprisingly low? Numbers from animal somatic tissues are in this range as well – for example, Harrison et al. found just one female-biased gene using similar fold-change thresholds in the spleen, and no male-biased genes. Based on this, I would suggest the title for this section is somewhat misleading, as perhaps there are someone more SBGs than expected based on studies in comparable tissues.

We changed the title of this section to "Levels of SBGE in *Leucadendron* leaves span the entire range observed in other dioecious plants". We also now refer to the single sex-biased gene in bird spleen, and explain that the contrast between animals and plants is that the former sometimes, but not always, may show much higher SBGE in non-reproductive tissue, depending on the tissue (Naqvi et al. 2019).

5. Line 216. This bit illustrates how the comparison to Harrison et al. (2015) seems unnecessarily contradictory. Yes, it is true that in Harrison et al. gonad samples clustered first by sex, then by species, but they also reported that when they clustered expression of spleen data, it clustered by species, then sex, just like the study in review here. This underlines the fact that the leaf tissue in question here is comparable to animal somatic tissue, not gonad. The authors need to be very careful and not over-emphasize differences that are in fact entirely concordant.

We have rephrased this section for clarity, and we have attempted throughout the manuscript to avoid claims and statements of contradiction between our paper and Harrison et al.'s study. Yes, leaf tissue is more comparable to animal somatic tissue than to animal gonads – but we do not think that this fact invalidates a comparison between the two studies. We expect that transcriptomes of any tissue, be they reproductive or non-reproductive, could in principle respond to sex-specific selection in a way similar to morphology (i.e., both may evolve sexual dimorphism). Of course, it is more plausible that gonads or inflorescences are affected by sex-specific selection than non-reproductive tissues (wattles, feathers, leaves, etc), but they are not expected to be the exclusive targets. In our understanding, the central point of Harrison et al's argument is that SBGE in gonads reflects the intensity of sexual selection because it correlates positively with known phenotypic indicators of sexual selection. We hypothesised that SBGE in leaf transcriptomes could correlate positively with leaf sexual dimorphism. Unfortunately, the link between vegetative sexual dimorphism and sex-specific selection in *Leucadendron* (and in plants in general) is not as well investigated and accepted as the link between sexual dimorphism and sexual selection in birds, but sex-specific selection remains the most plausible explanation for differences in leaf size. We found that there is no such correlation as in the birds, and this means, as we point out in the Conclusion, that in *Leucadendron* leaves, gene expression and morphology are not driven by the same selection pressures, or they do not respond to it similarly. We believe that the amended Introduction clarifies the role that sexual selection (or simply different selection on males and females) has probably played in the evolution of the strikingly different leaf morphology between the sexes in *Leucadendron*.

We agree that our result, that samples of plant leaf gene expression cluster by species before sex, is concordant with Harrison et al.'s result for spleen, and other reports from animal somatic tissues. In the previous version of the manuscript, we simply wished to point out that there is a notable difference in this clustering analysis between animal gonads and animal somatic tissue, but have now removed that confusing sentence as it is not essential here.

6. Delta-X comparison. It is important to differentiate increases from decreases in expression in delta- comparisons, because the loss (or very low expression and functional loss) of expression for a gene is expected under completely relaxed selection, and this would give a signature of positive selection under Delta-X (large change, little variation).

We report that expression changes (shifts) towards sex-bias are depleted in signatures of adaptation (high delta-x) relative to unbiased changes. The reviewer suggests that we may falsely infer adaptive expression shifts (i.e., high delta-x) for genes that have nearly or completely lost expression, because such genes would have high delta-x values, even if the loss of expression was due to completely relaxed selection. It is not clear how this bias would impact our finding, because in principle it should affect both sex-biased and unbiased changes similarly.

To investigate this further, we partitioned the increases and decreases, asking whether the pattern of depletion of high-delta-x among changes towards sex-bias is observed in both directions of change. These results are presented in detail as a new section of Appendix 2: "Robustness of inference of the proportion of adaptive shifts in gene expression (delta-x)". We found that the depletion of high-delta-x is more pronounced among expression decreases, but the trend is consistent among both increases and decreases in expression. Therefore, we believe it is justified not to distinguish decreases and increases in this case, and we have not changed the main text results.

Moreover, the way the authors have set up their Delta-X measure (similar to the Ometto et al. 2011 formula) will result in a conflation between highly-expressed genes and high Delta-X. This is a problem in the formulation used in the Ometto et al. (2011) analysis, and why Moghadam et al. (2012) used a somewhat more complicated formula which corrects for expression level. If the authors would like to use this formula for gene expression evolution, they should make sure that there is no relationship between expression level and Delta-X value.

We investigated whether this could be a problem in our data with absolute delta-X values (i.e., not distinguishing expression decreases and increases). Indeed, we find a slight positive correlation (Spearman's rho ~0.12) between mean expression level and our version of delta-x, and this is nearly the same with Moghadam's version of delta-x if applied to our data. However, if using Ometto's version, we find a much stronger correlation (rho = 0.62), as implied by the reviewer. Below are all three formulas, simplified to omit the sample size correction factors used by Moghadam et al. and Ometto et al:

Zemp / this study:Dx = meanFOCAL−meanREFSDFOCAL

Moghadam et al. 2012:Dx = (meanFOCAL−meanREF) / meanFOCAL RANGEFOCAL /  meanFOCAL

Ometto et al. 2011:Dx = meanFOCAL−meanREF SDFOCAL /  meanFOCAL

In their delta-X formula, Moghadam et al. (2012) divide both the numerator (divergence) as well as the denominator (diversity) by the mean expression level. As these terms cancel out, our formula and that of Moghadam are the same except that our diversity measure (denominator) is the standard deviation (SD) and not the sample range. Sample range as well as sample standard deviation are generally strongly positively correlated to sample mean. Therefore, we think that neither our formula nor that of Moghadam conflates high expression and high delta-X, but Ometto's formula clearly does. The key difference between our formula and that of Ometto is that for their denominator, Ometto chose the sample standard deviation divided by the sample mean (the coefficient of variation, or relative standard deviation). We and Moghadam use just the simple standard deviation or the range, respectively, which preserve the scale, allowing to "correct" for the scale (i.e., the mean expression level). We hope to have shown convincingly that our formula and that of Moghadam for delta-x are equivalent in regard to correction for the scale (mean expression), and we have made no changes to the text.

7. Table 1. I am unsure how the authors reached some of these predictions. For example, I do not understand why adaptive evolution would result in greater sequence and expression evolution than non-adaptive scenarios. One could hypothesize that very few codon changes would be adaptive, leading to lower dN/dS than relaxed constraint. Similarly, if gene expression is under selection, small changes might be expected under adaptive scenarios while relaxed constraint could lead to large changes with no effect. Similarly, dN/dS would be expected to be very high in genes under relaxed constraint, and likely higher than adaptive selection. The delta-X prediction would only be true for genes experiencing increased expression, the opposite is true to decreased expression – see my comment about Delta-X above.

We removed this table, as suggested by the editor, and explain our now better informed hypotheses in the text.

8. Why was the sequence polymorphism analysis not done on all species? This bit should also be discussed more thoroughly in the conclusions.

The power of the polyDFE method is dependent on the size of the site frequency spectra used as input. The smallest dataset considered by Tataru et al. (2019, *Bioinformatics*) had 0.89 Mbp, and already yielded substantially greater variation in the estimated parameters than their typical datasets with >> 1 Mbp. Among our sample of species, *L. dubium* contained the highest number of SBGs (about 140 of male- respectively female-biased genes), and these SBGs yielded site frequency spectra (SFS) with sizes between 32-73 kb, while the site frequency spectrum of unbiased genes was about 6 Mb long. Confidence Intervals (CIs) for unbiased genes (about 6 Mbp of data) spanned about 5-10% of the potential parameter space, whereas the CIs for the same parameters for SBGs spanned on average 30%, and up to 77%, of the potential parameter space (see the new DFE Figure, Figure 5—figure supplement 1). We thus report these results with a note of caution. Given this suboptimal analysis for the species with the greatest number of SBGs (largest SFS datasets), we refrained from repeating it in other species, which contained far fewer SBGs (about 100 at most). However, for comparison we have added an estimation of the DFE of the roughly 280 *L. dubium* SBGs in a second species, *L. ericifolium*. This provided new insights, although caution is again advised due to the great uncertainties.

We have added the following sentence to our Conclusion: "Standing population genetic variation in one species in our sample hints at the possibility that the gain of female-biased expression is associated with a relaxation of purifying selection."